# MeanCache: From Instantaneous to Average Velocity for Accelerating Flow Matching Inference

**Huanlin Gao**[1,2], **Ping Chen**[1,2], **Fuyuan Shi**[1,2], **Ruijia Wu**[2], **Yantao Li**[1,2,3], **Qiang Hui**[1,2]
**Yuren You**[2], **Ting Lu**[1,2], **Chao Tan**[1,2], **Shaoan Zhao**[1,2], **Zhaoxiang Liu**[1,2]
**Fang Zhao**[1,2*], **Kai Wang**[1,2], **Shiguo Lian**[1,2*]

[1]Data Science & Artificial Intelligence Research Institute, China Unicom
[2]Unicom Data Intelligence, China Unicom
[3]National Key Laboratory for Novel Software Technology, Nanjing University
Code: UnicomAI/MeanCache

## Abstract

We present **MeanCache**, a training-free caching framework for efficient Flow Matching inference. Existing caching methods reduce redundant computation but typically rely on instantaneous velocity information (e.g., feature caching), which often leads to severe trajectory deviations and error accumulation under high acceleration ratios. MeanCache introduces an average-velocity perspective: by leveraging cached Jacobian–vector products (JVP) to construct interval average velocities from instantaneous velocities, it effectively mitigates local error accumulation. To further improve cache timing and JVP reuse stability, we develop a trajectory-stability scheduling strategy as a practical tool, employing a Peak-Suppressed Shortest Path under budget constraints to determine the schedule. Experiments on FLUX.1, Qwen-Image, and HunyuanVideo demonstrate that MeanCache achieves $4.12\times$, $4.56\times$, and $3.59\times$ acceleration, respectively, while consistently outperforming state-of-the-art caching baselines in generation quality. We believe this simple yet effective approach provides a new perspective for Flow Matching inference and will inspire further exploration of stability-driven acceleration in commercial-scale generative models.

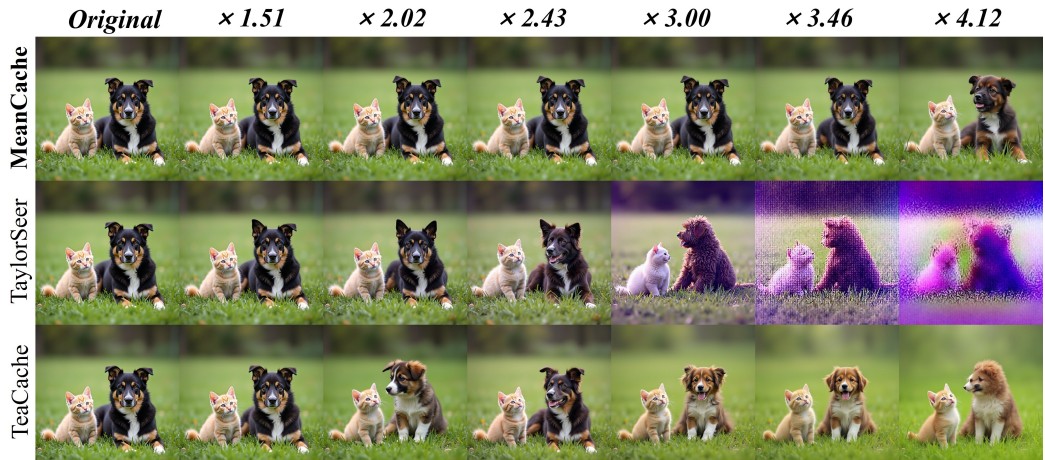

Figure 1: Visualization of images generated by different methods on **FLUX.1[dev]** under varying acceleration ratios.

---

*Corresponding author
{gaohl51, zhaof50, liansg}@chinaunicom.cn

## 1 INTRODUCTION

Flow Matching (Lipman et al., 2023; Albergo & Vanden-Eijnden, 2022) has recently demonstrated remarkable progress across image (Wu et al., 2025), video (Zheng et al., 2024b; Kong et al., 2024), and multi-modal generation tasks (Hung et al., 2024). By modeling instantaneous velocity fields to learn continuous transport paths, it offers a concise and effective paradigm for generative modeling. However, in commercial-scale models such as FLUX.1 (Labs, 2024), Qwen-Image (Wu et al., 2025), and HunyuanVideo (Kong et al., 2024), the large memory footprint, heavy per-step computational cost, and long inference latency significantly hinder its applicability in interactive or resource-constrained scenarios.

Traditional acceleration methods, such as distillation (Salimans & Ho, 2022; Kim et al., 2023; Sauer et al., 2024b), pruning (Han et al., 2015), and quantization (Li et al., 2023b), usually rely on architecture modification and large-scale retraining. In contrast, caching-based methods (Ma et al., 2023a) offer a lightweight, training-free alternative. By reusing intermediate representations from selected timesteps, they reduce redundant computation and accelerate sampling. However, at high acceleration ratios, these methods often suffer from severe **error accumulation**: interval states reconstructed solely from instantaneous velocity or feature information amplify local deviations, causing the trajectory to drift away from the true path. As shown in Fig. 2 (left), instantaneous velocities fluctuate sharply along the denoising trajectory, making them unstable for reuse, whereas interval average velocities are much smoother and thus more stable for reconstruction.

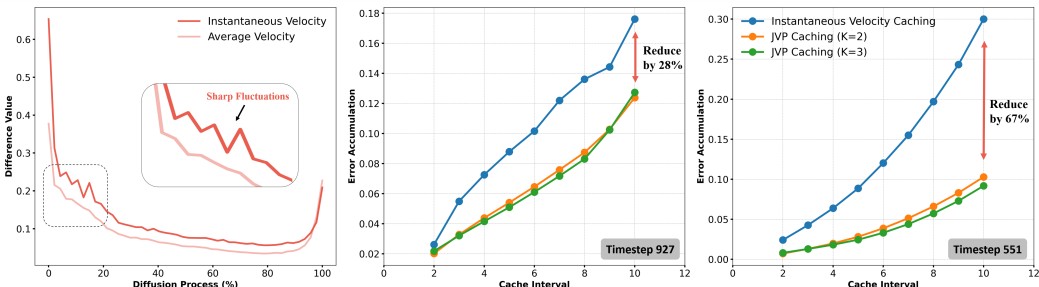

Figure 2: **Instantaneous vs. Average Velocity and JVP Caching. (Left)** Along the original trajectory, instantaneous velocity shows sharp fluctuations, while average velocity is much smoother. **(Middle)** At timestep 927, JVP Caching reduces error accumulation, though its effectiveness depends on the cache interval and hyperparameter $K$. **(Right)** At timestep 551, it achieves stronger error mitigation, showing that effectiveness varies across timesteps. Both middle and right figures are under the single-cache setting on the original trajectory.

This observation is consistent with the objective of Flow Matching, which encourages trajectories to satisfy linear characteristics. Under fixed input conditions, an ideal trajectory approximates linear interpolation between sample and noise; the more linear the trajectory, the more stable and higher-quality the generation results. Recent work such as MeanFlow (Geng et al., 2025), further demonstrates that modeling and leveraging average velocity can significantly improve trajectory stability, underscoring the potential of the average-velocity domain for more robust generation.

Motivated by this insight, we propose **MeanCache**, a training-free caching paradigm that operates in the average-velocity domain rather than relying solely on instantaneous velocities. The key idea is to construct interval average velocities from instantaneous ones under a limited budget, ensuring trajectory stability. MeanCache has two components. First, interval average velocities are approximated using cached Jacobian–vector products (JVP), yielding smoother and more stable guidance signals that help mitigate local error accumulation. As shown in Fig. 2 (middle, right), JVP caching reduces errors at timesteps 927 and 551; however, its benefit varies with the timestep, cache interval, and hyperparameters, indicating that fixed caching rules are insufficient. Second, we develop a trajectory-stability scheduling strategy as a practical tool. Inspired by the graph-based modeling idea in ShortDF (Chen et al., 2025) and by trajectory drift mitigation (Lin et al., 2019) in sequence processing (Wang et al., 2019), timesteps are represented as nodes, deviations of average velocity under JVP caching define edge weights, and a budget-constrained shortest-path search determines cache placement. This scheduling tool systematically improves cache timing and JVP reuse stability without retraining.

The main contributions of this work are summarized as follows:

- **Average-Velocity Perspective on Caching.** We introduce MeanCache, which redefines the caching problem from an instantaneous velocity view to the average-velocity domain, offering a simpler and more stable perspective for high-acceleration generative modeling.
- **Trajectory-Stability Scheduling Strategy.** We develop a scheduling tool that scores timesteps by JVP-based stability deviation and uses a budget-constrained shortest-path search for cache placement, improving timing and reuse stability without retraining.
- **Outstanding Performance.** MeanCache maintains generation quality under high acceleration while significantly reducing inference cost. Compared to state-of-the-art caching baselines, experiments on FLUX.1, Qwen-Image, and HunyuanVideo show speedups of $4.12\times$, $4.56\times$, and $3.59\times$, respectively. Moreover, MeanCache consistently delivers higher generation quality across different acceleration ratios (Fig. 1), highlighting its acceleration potential on commercial-scale generative models.

## 2 METHODOLOGY

### 2.1 PRELIMINARIES

**Flow Matching and MeanFlow.** Flow Matching (Lipman et al., 2023; Albergo & Vanden-Eijnden, 2022) constructs continuous transport paths between a noise distribution $\pi_1$ and a data distribution $\pi_0$ via velocity fields, typically defined by linear interpolation $x_t = (1 - t)x_0 + tx_1$, $t \in [0, 1]$. This leads to the ODE $dx_t = (x_0 - x_1)dt$. Since $x_0$ is unknown during generation, a neural network $v_\theta(x_t, t)$ is trained to predict the instantaneous velocity, yielding the dynamics

$$d\hat{x}_t = v_\theta(x_t, t) \, dt. \tag{1}$$

Where $\hat{x}_t$ denotes the trajectory point predicted by the neural ODE. Building on this formulation, MeanFlow (Geng et al., 2025) offers a new perspective by modeling the average velocity over the interval $[s, t]$, defined as

$$u(z_s, t, s) = \frac{1}{s-t} \int_t^s v(z_\tau, \tau) \, d\tau, \tag{2}$$

Furthermore, the MeanFlow Identity provides a theoretical bridge between instantaneous and average velocity:

$$v(z_s, s) = u(z_s, t, s) + (s - t)\frac{d}{ds}u(z_s, t, s). \tag{3}$$

In this identity, the derivative term $\frac{d}{ds}u$ can be expressed as a **Jacobian–Vector Product (JVP)**, where the Jacobian of $u$ with respect to $(z, s)$ is contracted with the tangent vector $[v(z_s, s), 1]$. Observing this formulation, JVP can be regarded as a computational bridge that directly connects instantaneous velocity and average velocity.

**Feature Caching in Diffusion Models.** Cache, as a training-free acceleration method, speeds up the denoising process by storing intermediate features and reusing them across adjacent timesteps. In particular, the reuse strategy directly substitutes cached features from a previous step:

$$\mathcal{F}(x_{t-k}^l) := \mathcal{F}(x_t^l), \quad \forall k \in [1, N - 1], \tag{4}$$

where $\mathcal{F}$ denotes the feature extraction function, and $l$ is the layer index. This approach avoids redundant computations and yields up to $(N - 1)\times$ theoretical speedup. Nevertheless, two major limitations remain:

(i) **Error Accumulation**: Ignoring the temporal dynamics of features leads to exponential error accumulation as $k$ increases. Although cache-then-forecast methods have been proposed recently (Liu et al., 2025), the extrapolated features still exhibit

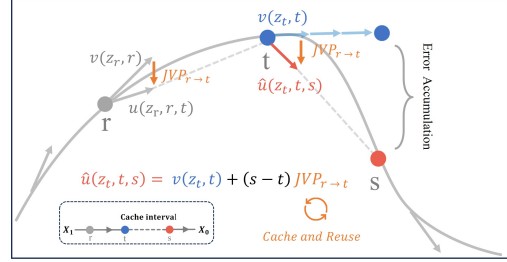

Figure 3: **From Instantaneous to Average Velocity.** Directly caching the instantaneous velocity $v(z_t, t)$ over $[t, s]$ easily leads to trajectory drift and error accumulation, whereas the average velocity $u(z_t, t, s)$ accurately reaches the target $s$. MeanCache introduces a prior timestep $r$ and reuses $\text{JVP}_{r \to t}$ to estimate the average velocity $\hat{u}(z_t, t, s)$, thereby correcting the trajectory and effectively mitigating error accumulation.

significant deviations from the true trajectories,
limiting acceleration in generative tasks.

(ii) **When to Cache**: Existing methods for deciding when to cache rely on fixed intervals (Ma et al., 2023a) or manually tuned threshold-based strategies (Liu et al., 2024; Bu et al., 2025), but these approaches cause significant degradation in generative quality under high acceleration ratios.

## 2.2 Instantaneous to Average Velocity Transformation

Traditional feature caching methods operate in the instantaneous-velocity domain, where velocity varies continuously along the trajectory and inevitably accumulates errors (Fig. 2). Inspired by the MeanFlow Identity, we instead reformulate caching in the average-velocity domain. As shown in Fig. 3, transforming the instantaneous velocity $v(z_t, t)$ into the average velocity $u(z_t, t, s)$ over the interval $[s, t]$ can, in principle, correct the trajectory accurately and eliminate accumulated errors. MeanCache builds on this perspective by formally deriving and practically approximating $u(z_t, t, s)$.

The MeanFlow Identity in Eq. 3 characterizes the instantaneous velocity only at the endpoint $s$, leaving the starting point $t$ unspecified. To close this gap, we derive an analogous relation at $t$ (see A.1 for details):

$$v(z_t, t) = u(z_t, t, s) - (s - t) \frac{d}{dt} u(z_t, t, s).$$ (5)

Here, the derivative term $\frac{d}{dt} u(z_t, t, s)$ can be expressed as a Jacobian–Vector Product (JVP). Since the exact JVP is unavailable during inference, we approximate it using cached values from earlier steps. This yields the following estimate for the average velocity:

$$\widehat{u}(z_t, t, s) := v(z_t, t) + (s - t) \widehat{\text{JVP}},$$ (6)

where $\widehat{\text{JVP}}$ denotes an approximation to the total derivative $\frac{d}{dt} u(z_t, t, s)$, and $\widehat{u}(z_t, t, s)$ represents the estimated average velocity.

## 2.3 JVP-based Cache Construction

To construct a practical cache estimator, we extend the start-point identity by introducing a reference point $r$ preceding $t$, with $r > t > s$. Intuitively, $r$ serves as an earlier cached state that helps approximate the JVP between $t$ and $s$, as illustrated in Fig. 3. Applying the start-anchored identity on the interval $[t, r]$ and rearranging gives:

$$\widehat{\text{JVP}} = \frac{d}{dr} u(z_r, r, t) = \frac{u(z_r, r, t) - v(z_r, r)}{t - r}.$$ (7)

Using the displacement form of the average velocity on $[r, t]$,

$$u(z_r, r, t) = \frac{z_t - z_r}{t - r},$$ (8)

we obtain the fully cacheable estimator:

$$\widehat{\text{JVP}} = \frac{z_t - z_r - (t - r) v(z_r, r)}{(t - r)^2}.$$ (9)

Plugging this into the start-point identity yields the predicted average velocity:

$$\boxed{\widehat{u}(z_t, t, s) = v(z_t, t) + (s - t) \frac{z_t - z_r - (t - r) v(z_r, r)}{(t - r)^2}}.$$ (10)

We denote by $K$ the number of discrete timesteps between $r$ and $t$ in the original trajectory. The final estimator is:

$$\widehat{u}(z_t, t, s) = \begin{cases} v(z_t, t) + (s - t) \widehat{\text{JVP}}_K, & K > 1, \\ v(z_t, t), & K = 1, \end{cases}$$ (11)

where larger $K$ corresponds to reusing cached information over a longer interval, while $K = 1$ reduces the average velocity to the instantaneous one. In the original denoising trajectory (e.g., 50 steps), $z_r$, $z_t$, and $z_s$ are available, so exact average velocities and JVP can be computed. Under caching, however, $\text{JVP}_{t \to s}$ is unavailable and must be approximated using $\text{JVP}_{r \to t}$. Thus, the choice of $K$ is critical: it specifies the span of the preceding segment ($r \to t$) used to approximate $\text{JVP}_{t \to s}$, balancing approximation error and stability. This trade-off motivates the need for a principled scheduling strategy.

## 2.4 TRAJECTORY-STABILITY SCHEDULING

Although JVP-based corrections mitigate local error accumulation, two key challenges remain: determining when to cache and how to select the cache span $K$. Empirically, while latent values vary across samples (e.g., different prompts and seeds), their relative changes at fixed timesteps are highly consistent. This stability, also observed in adaptive schemes such as TeaCache (Liu et al., 2024), indicates that caching decisions can be guided by a precomputed stability map rather than fixed heuristics.

**Stability Map via Graph Representation.** Specifically, we define the error from $t$ to $s$ as the deviation between the true average velocity and its cached approximation:

$$\mathcal{L}_K(t, s) = \| u(z_t, t, s) - \widehat{u}(z_t, t, s) \|. \tag{12}$$

Expanding the cached estimator $\widehat{u}(z_t, t, s)$ with JVP correction gives:

$$\mathcal{L}_K(t, s) = \frac{1}{N} \left\| u(z_t, t, s) - v(z_t, t) - (s - t) \widehat{\text{JVP}}_K \right\|_1, \tag{13}$$

To support trajectory-stability scheduling, we use a graph representation as a practical tool to organize stability costs and possible transitions. Specifically, for convenience, this can be represented as a graph $\mathcal{G} = (\mathcal{V}, \mathcal{E})$, where nodes $\mathcal{V}$ correspond to timesteps in the denoising process and edges $\mathcal{E}$ are directed connections ($t \to s$) with $t > s$, each representing a potential caching transition. Each edge is assigned a weight:

$$\mathcal{E}_K(t \to s) = \mathcal{L}_K(t, s), \qquad t, s \in \mathcal{V}. \tag{14}$$

where $\mathcal{L}_K(t, s)$ is the error between predicted and true average velocities under cache span $K$. Since multiple cache spans may connect the same node pair, $\mathcal{G}$ is naturally modeled as a multigraph.

**Peak-Suppressed Shortest Path.** Given a Multigraph with error-weighted edges, the scheduling problem can be conveniently solved via a constrained shortest-path search. A challenge under small budgets is that the solution may concentrate error into a few edges, leading to large error spikes. To address this, we adopt a peak-suppressed objective that penalizes high-error edges via a power-weighted path cost. The optimization problem is:

$$\pi^\star = \arg\min_{\pi \in \mathcal{P}(T, 0)} \sum_{e \in \pi} \mathcal{C}(e)^\gamma \quad \text{s.t.} \quad |\pi| \leq \mathcal{B} \leq T, \tag{15}$$

where $\mathcal{P}(T, 0)$ is the set of feasible multi-edge paths from the start node $T$ to the end node $0$, $\mathcal{C}(e)$ is the error cost of edge $e$, $\gamma \geq 1$ is the peak-suppression parameter ($\gamma = 1$ recovers the standard shortest path), and $|\pi|$ is the path length. This peak-suppressed shortest-path problem can be solved efficiently via dynamic programming. The budget $\mathcal{B}$ acts as a constraint on the original path cost and directly controls the acceleration ratio.

## 3 EXPERIMENTS

### 3.1 EXPERIMENTAL SETUP

**Baselines and Compared Methods.** We evaluate our method on representative diffusion-based generative models: FLUX.1 [dev] (Labs, 2024), Qwen-Image (Wu et al., 2025), and Hunyuan-Video (Kong et al., 2024). Baselines include TeaCache (Liu et al., 2024), DBCache (vipshop.com, 2025), DiCache (Bu et al., 2025), ToCa (Zou et al., 2024a), DuCa (Zou et al., 2024b), and TaylorSeer (Liu et al., 2025). Among them, TeaCache (Liu et al., 2024) and TaylorSeer (Liu et al., 2025) are two of the most representative mainstream approaches, spanning both text-to-image and text-to-video generation tasks.

Table 1: Quantitative comparison in text-to-image generation on **FLUX.1 [dev] and Qwen-Image**.

| Method | Acceleration | | | Visual Quality | | | | |
|---|---|---|---|---|---|---|---|---|
| | FLOPs(T) ↓ | Latency(s) ↓ | Speed ↑ | Image Reward ↑ | CLIP Score ↑ | LPIPS ↓ | SSIM ↑ | PSNR ↑ |
| **FLUX.1 [dev]** 1024×1024 | | | | | | | | |
| **Original: 50 steps** | 3734.56 | 11.57 | – | 1.033 | 31.229 | – | – | – |
| 60% steps | 2246.87 | 7.01 | 1.65× | 0.984 | 31.242 | 0.217 | 0.808 | 20.256 |
| 30% steps | 1131.10 | 3.60 | 3.21× | 0.880 | 30.832 | 0.399 | 0.682 | 15.798 |
| TeaCache ($l = 0.25$) | 1949.73 | 4.62 | 2.50× | 0.960 | 31.145 | 0.338 | 0.721 | 17.286 |
| DiCache ($\delta = 0.8$) | 1032.51 | 4.32 | 2.68× | 0.675 | 30.814 | 0.416 | 0.717 | 21.268 |
| TaylorSeer ($\mathcal{N} = 6, O = 2$) | 760.08 | 4.24 | 2.74× | 0.971 | **31.310** | 0.415 | 0.663 | 16.278 |
| TaylorSeer ($\mathcal{N} = 6, O = 1$) | 760.08 | 4.06 | 2.85× | 0.961 | 31.191 | 0.419 | 0.660 | 15.831 |
| **MeanCache ($\mathcal{B} = 15$)** | 1131.10 | **3.98** | **2.91×** | **1.010** | 31.244 | **0.142** | **0.870** | **24.834** |
| TeaCache ($l = 1.5$)† | 536.73 | 3.16 | 3.66× | 0.717 | 30.696 | 0.504 | 0.624 | 15.010 |
| DiCache ($\delta = 2.0$)† | 958.15 | 3.14 | 3.68× | -0.652 | 27.613 | 0.586 | 0.588 | 17.446 |
| TaylorSeer ($\mathcal{N} = 20, O = 1$)† | 388.27 | 3.10 | 3.73× | -0.727 | 24.412 | 0.798 | 0.443 | 11.219 |
| **MeanCache ($\mathcal{B} = 10$)** | 759.18 | **2.81** | **4.12×** | **0.993** | **31.323** | **0.272** | **0.761** | **19.425** |
| **Qwen-Image** 1664×928 | | | | | | | | |
| **Original: 50 steps** | 10928.60 | 32.68 | 1.00× | 1.180 | 33.626 | – | – | – |
| 30% steps | 3291.75 | 9.86 | 3.31× | 1.128 | 33.026 | 0.363 | 0.727 | 15.826 |
| TeaCache ($l = 0.6$) | 5481.27 | 18.52 | 1.76× | 1.087 | 32.598 | 0.416 | 0.698 | 14.902 |
| DBCache ($r = 0.6$) | 2703.00 | 11.92 | 2.74× | 1.016 | 33.435 | 0.298 | 0.825 | 22.221 |
| **MeanCache ($\mathcal{B} = 15$)** | 3291.75 | **11.45** | **2.85×** | **1.159** | **33.636** | **0.075** | **0.938** | **27.663** |
| DBCache ($r = 1.5$)† | 2070.26 | 9.57 | 3.41× | -2.059 | 15.499 | 0.889 | 0.129 | 5.559 |
| DBCache + Taylorseer ($r = 1.5, O = 4$)† | 2070.26 | 9.70 | 3.37× | -0.227 | 29.753 | 0.625 | 0.646 | 16.574 |
| **MeanCache ($\mathcal{B} = 13$)** | 2855.35 | 9.09 | 3.60× | **1.147** | **33.799** | **0.113** | **0.907** | **24.802** |
| **MeanCache ($\mathcal{B} = 10$)** | 2200.77 | **7.16** | **4.56×** | 1.142 | 33.621 | 0.236 | 0.815 | 18.983 |

- † Methods exhibit significant degradation in Image Reward, leading to severe deterioration in image quality.

**Metrics.** For a fair comparison, we evaluate both efficiency and quality. Efficiency is measured by FLOPs and latency, while quality is assessed with task-specific and reconstruction metrics. For text-to-image generation, we follow the standard DrawBench (Saharia et al., 2022) protocol and report ImageReward (Xu et al., 2023) and CLIP Score (Hessel et al., 2021) to evaluate perceptual quality and text–image alignment. For text-to-video generation, we adopt VBench (Huang et al., 2024) to capture human preference on generated videos. In addition, for both tasks, we report LPIPS (Zhang et al., 2018) (perceptual similarity), SSIM (Wang & Bovik, 2002) (structural consistency), and PSNR (pixel-level accuracy) to quantify potential degradation in content and fidelity introduced by acceleration.

**Implementation details.** Experiments are conducted on NVIDIA H100 GPUs using PyTorch. To construct the multigraph, we sample 50 prompts (10 per attribute) from T2V-CompBench (Sun et al., 2024), following standard practice (Sun et al., 2024; Liu et al., 2024). This procedure is applied consistently across both text-to-image and text-to-video experiments, even though the dataset was originally not designed for text-to-image generation. Sampling is repeated 5 times with different seeds, and results are averaged to reduce bias. For all experiments, FlashAttention (Dao et al., 2022) is enabled by default to accelerate attention computation. Notably, since TaylorSeer encounters out-of-memory (OOM) issues under HunyuanVideo, we uniformly adopt the cpu-offload setting to ensure fair comparison.

### 3.2 TEXT-TO-IMAGE GENERATION.

As shown in Table 1, MeanCache achieves clear quantitative gains on two advanced text-to-image models, FLUX and Qwen-Image. We use ImageReward (Xu et al., 2023) and CLIP Score (Hessel et al., 2021) as perceptual metrics, and reconstruction metrics to measure content and detail preservation. On FLUX, at 2.91× acceleration, MeanCache surpasses TaylorSeer and TeaCache in both image quality and detail preservation, and remains robust at higher ratios. Even at 4.12×, where competitors collapse in quality, our method still attains an ImageReward Score↑ of 0.993 and an LPIPS↓ of 0.272. On Qwen-Image (1664×928 resolution), MeanCache likewise im-

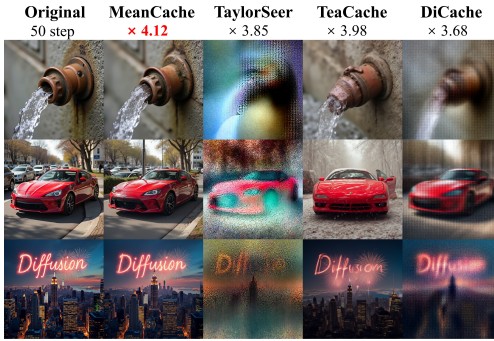

Figure 4: Comparison of different methods at high acceleration ratios on **FLUX.1[dev]**.

Table 2: Quantitative comparison in text-to-video generation on **HunyuanVideo** †.

| Method | Acceleration | | Visual Quality | | | |
|---|---|---|---|---|---|---|
| *HunyuanVideo* | **Latency(s) ↓** | **Speed ↑** | **VBench ↑** | **LPIPS ↓** | **SSIM ↑** | **PSNR ↑** |
| **Original: 50 steps** | 105.92 | 1.00× | 80.39% | – | – | – |
| 30% steps | 39.53 | 2.68× | 79.84% | 0.381 | 0.659 | 17.335 |
| ToCa ($\mathcal{N} = 5$) | 36.17 | 2.93× | 79.51% | 0.454 | 0.590 | 15.765 |
| Duca ($\mathcal{N} = 5$) | 34.32 | 3.09× | 79.54% | 0.454 | 0.595 | 15.807 |
| DiCache ($\delta = 0.8$) | 33.76 | 3.11× | 74.09% | 0.382 | 0.701 | 22.053 |
| TaylorSeer ($\mathcal{N} = 5, O = 1$) | 34.95 | 3.03× | 79.95% | 0.428 | 0.603 | 16.026 |
| Teacache ($l = 0.33$) | 34.06 | 3.11× | **80.02%** | 0.363 | 0.651 | 17.957 |
| **MeanCache ($\mathcal{B} = 12$)** | **33.05** | **3.21×** | 80.01% | **0.176** | **0.809** | **24.002** |
| DiCache ($\delta = 3.0$) | 31.81 | 3.33× | 70.86% | 0.583 | 0.490 | 19.098 |
| Teacache ($l = 0.39$) | 31.86 | 3.32× | 79.75% | 0.396 | 0.631 | 17.382 |
| TaylorSeer ($\mathcal{N} = 7, O = 1$) | 31.50 | 3.36× | 79.76% | 0.480 | 0.595 | 15.444 |
| **MeanCache ($\mathcal{B} = 10$)** | **29.48** | **3.59×** | **80.08%** | 0.269 | 0.732 | 20.464 |

• † TaylorSeer may encounter OOM; for fairness, all methods are run with CPU-offload enabled.

proves both quality and speed, reaching an LPIPS↓ of 0.075 at 2.85× acceleration, indicating near-lossless sampling. As shown in Fig. 4, on FLUX.1 [dev], when the acceleration exceeds 3.5×, baseline methods suffer from severe blurring, detail loss, and structural distortions, whereas MeanCache consistently preserves perceptual quality and fidelity close to the original outputs. On Qwen-Image, MeanCache also demonstrates strong robustness under high acceleration ratios, outperforming other baselines as illustrated in Fig. 9.

### 3.3 TEXT-TO-VIDEO GENERATION.

On the HunyuanVideo, Table 2 demonstrates the acceleration performance of MeanCache across VBench and three reconstruction metrics. With a speedup of 3.42×, our method significantly outperforms the main competitors, achieving 0.809 in SSIM↑ and 24.002 in PSNR↑. Performance continues to improve with a further 3.97× speedup, while maintaining a VBench score of 80.08%. In terms of content preservation, our method effectively preserves both the content and intricate details of the original video, surpassing all baseline methods in this regard. As shown in Fig. 5, when the acceleration exceeds 3.0×, baseline methods suffer from visual degradation or blurring, whereas MeanCache maintains superior video quality and fidelity to the original videos.

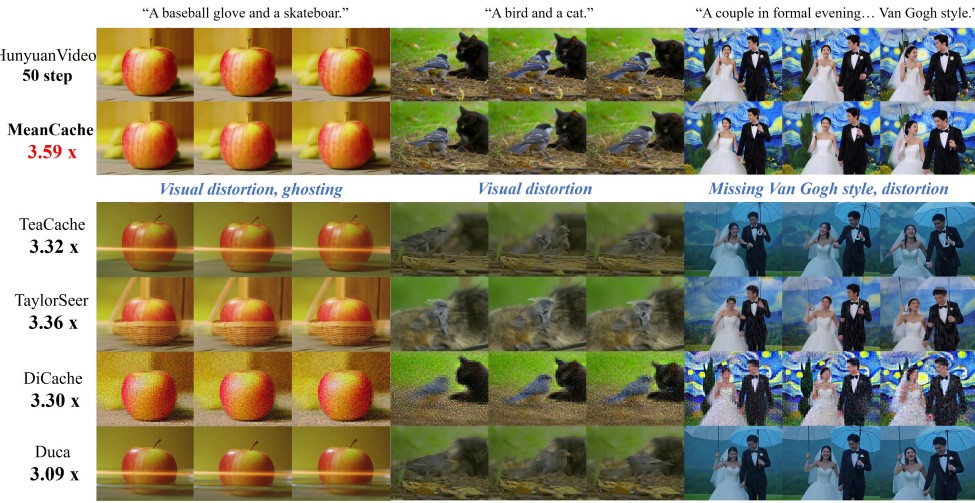

Figure 5: Comparison of different methods at high acceleration ratios on **HunyuanVideo**.

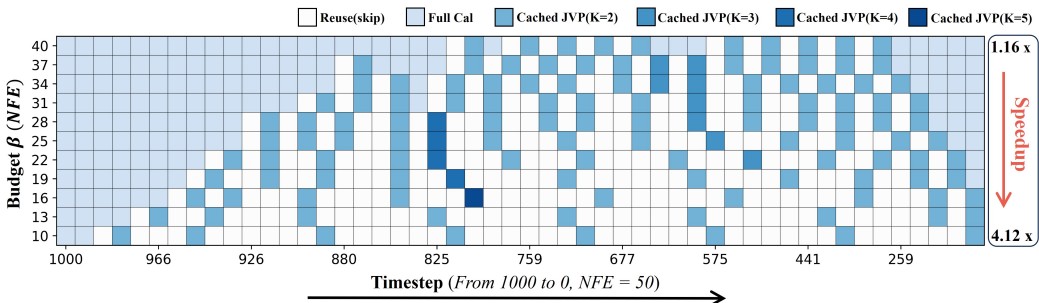

Figure 6: Shortest paths on a multigraph under different budgets $\mathcal{B}$ in FLUX.1[dev]

## 3.4 ABLATION STUDY

**Shortest Path Analysis.** Trajectory-Stability Scheduling is realized by constructing a multigraph and solving for its shortest paths. Given this representation, the shortest path under any step budget $\mathcal{B}$ can be obtained via edge-weighted optimization. The budget $\mathcal{B}$ (equivalent to the Number of Function Evaluations, NFE) directly controls the acceleration ratio: smaller values correspond to greater speedups. Figure 6 illustrates the shortest-path patterns on FLUX as $\mathcal{B}$ decreases from 40 to 10. The horizontal axis corresponds to the 50-step denoising trajectory, while the vertical axis indicates different budget levels. Darker cells represent larger cached JVP spans $K$, and gray cells indicate reuse (skips). This analysis reveals that early timesteps are crucial for denoising quality, whereas later timesteps, particularly in the latter half, contribute less and are more suitable for skipping. Moreover, the optimal JVP span $K$ is not fixed but depends jointly on the budget and timestep, underscoring the necessity of multigraph-based modeling for analyzing acceleration from the average-velocity perspective.

**Effect of Peak-Suppression Parameter** $\gamma$. The peak-suppression parameter, $\gamma$, controls the degree of peak suppression in the shortest path, effectively mitigating the concentration of error into a small number of edges. We selected a moderate budget size of $\mathcal{B} = 15$ and varied $\gamma$ within the range $[1, 5]$. The results, shown in Table 3, indicate that when $\gamma = 1$, the image quality metrics fail to reach optimal performance, suggesting the presence of error spikes within the shortest path. In contrast, when $\gamma = 5$, all evaluation metrics achieve their best performance, highlighting the effectiveness of peak suppression.

Table 3: Impact of peak-suppression parameter $\gamma$ on quality metrics.

| $\gamma$ | Reward↑ | CLIP↑ | LPIPS↓ | SSIM↑ | PSNR↑ |
|---|---|---|---|---|---|
| 1 | 1.0136 | 31.201 | 0.192 | 0.826 | 22.376 |
| 2 | 1.0072 | 31.195 | 0.148 | 0.860 | 24.147 |
| 3 | 1.0066 | 31.208 | 0.145 | 0.862 | 24.183 |
| 4 | **1.0179** | **31.291** | **0.135** | 0.869 | **24.569** |
| 5 | 1.0177 | 31.271 | 0.140 | **0.871** | 24.568 |

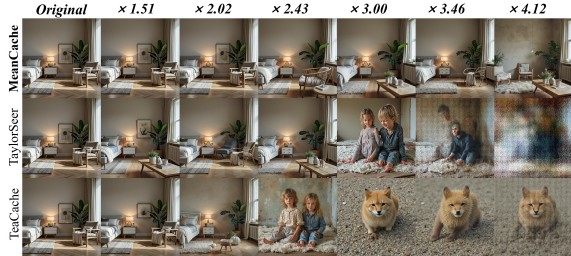

Figure 7: Content consistency under rare-word prompts *"Matutinal"* across acceleration ratios.

**Content Consistency** Content consistency before and after acceleration is a key criterion for evaluating acceleration methods. Rare words, due to their ambiguous semantics and infrequent usage, pose a stringent challenge for text-to-image generation. To assess consistency under acceleration, we compare MeanCache with two baselines, TaylorSeer (Liu et al., 2025) and TeaCache (Liu et al., 2024), using prompts containing rare words. As shown in Figure 7, all three methods maintain good consistency at low acceleration ratios ($< 2.43\times$). However, as the ratio increases, TaylorSeer and TeaCache exhibit severe content drift and quality degradation, whereas MeanCache preserves most of the original content and details even at a $4.12\times$ speedup.

## 4 RELATED WORK

**Diffusion Model Acceleration/Flow Models.** Diffusion models have achieved remarkable success across modalities, yet their iterative denoising procedure incurs high inference latency, making

acceleration a central challenge. A large body of work has therefore focused on reducing sampling steps. For example, DDIM (Song et al., 2020a) extends the original DDPM (Ho et al., 2020) to non-Markovian dynamics for faster sampling, while EDM (Karras et al., 2022) introduces principled design choices to improve efficiency. In parallel, advanced numerical solvers for SDEs/ODEs (Song et al., 2020b; Jolicoeur-Martineau et al., 2021; Lu et al., 2022; Chen et al., 2025) significantly improve the trade-off between accuracy and speed. Another line of work leverages knowledge distillation (Hinton et al., 2015), compressing multi-step trajectories into compact few-step models (Luo et al., 2023). Representative approaches include Progressive Distillation (Salimans & Ho, 2022), Consistency Distillation (Song et al., 2023; Kim et al., 2023; Geng et al., 2024; Wang et al., 2025; Zheng et al., 2024a), Adversarial Diffusion Distillation (Sauer et al., 2024b;a), and Score Distillation Sampling (Yin et al., 2024b;a). Orthogonal strategies such as quantization (Li et al., 2023b; So et al., 2023; Shang et al., 2023), pruning (Han et al., 2015; Ma et al., 2023b), system-level optimization (Liu et al., 2023), and parallelization frameworks (Zhao et al., 2024a; Li et al., 2024; Fang et al., 2024; Chen et al., 2024b) have also been explored to enhance efficiency.

Beyond these efforts, Flow Matching (Lipman et al., 2023; Albergo & Vanden-Eijnden, 2022) has emerged as a promising alternative. Unlike diffusion models (Song et al., 2020a;b) that rely on noise injection and SDE solvers, it learns velocity fields for distributional transformations and can be viewed as a continuous-time normalizing flow (Rezende & Mohamed, 2015). Extensions include Flow Map (Boffi et al., 2024) for integral displacements, Shortcut Models (Frans et al., 2025) for interval self-consistency, and Inductive Moment Matching (Zhou et al., 2025) for stochastic consistency. MeanFlow (Geng et al., 2025) further shifts the focus from instantaneous to average velocity, offering a new perspective on efficient generative modeling. Nevertheless, most methods still require heavy computation, large data, or complex engineering, limiting practical adoption.

**Cache in Diffusion Models.** Recently, caching strategies (Smith, 1982) have emerged as a promising retraining-free approach for accelerating diffusion inference (Wimbauer et al., 2023; Ma et al., 2024). The core idea is to reuse intermediate results from selected timesteps during sampling to reduce redundant computation (Selvaraju et al., 2024). Early attempts such as DeepCache (Ma et al., 2023a) accelerated the UNet backbone with handcrafted rules. Later, T-GATE (Zhang et al., 2024) and $\Delta$-DiT (Chen et al., 2024a) extended this idea to DiT architectures (Peebles & Xie, 2023), achieving significant speed-ups in image synthesis (Li et al., 2023a). With the breakthrough of Sora (OpenAI, 2024) in video generation, these techniques have also been extended to temporal domains. For instance, PAB (Zhao et al., 2024b) identified a U-shaped trajectory of attention differences across timesteps and proposed a cache-and-broadcast strategy. More recently, TaylorSeer (Liu et al., 2025) combined multi-step cached features in a Taylor-expansion-like manner to enhance feature reuse; TeaCache (Liu et al., 2024) exploits the correlation between timestep embeddings and model outputs, employing thresholding and polynomial fitting to guide its caching strategy. Di-Cache (Bu et al., 2025) enhances this idea with online shallow-layer probes, while DBCache (vip-shop.com, 2025) extends TeaCache's thresholding scheme to additional boundary blocks. LeM-iCa (Gao et al., 2025) proposes a global caching mechanism based on a DAG structure to accelerate video synthesis. While effective, these methods mainly adopt an instantaneous-velocity perspective, performing feature caching at the step-wise level. This view is inherently unstable (Fig. 2), often leading to trajectory drift and error accumulation under high acceleration ratios.

## 5 Discussion and Conclusion

In this work, we presented **MeanCache**, a lightweight and training-free caching framework for Flow Matching. By shifting the perspective from instantaneous to average velocities and combining JVP-based estimation with trajectory-stability scheduling, MeanCache effectively mitigates error accumulation and improves cache placement. Experiments on commercial-scale models demonstrate that it achieves significant acceleration while preserving high-fidelity generation. Beyond its empirical performance, MeanCache contributes a new perspective to caching methods: rather than reusing instantaneous quantities, it leverages average-velocity formulations as a more stable foundation. This not only enriches the design space of caching strategies, but also extends the applicability of average-based velocities ideas, such as those in MeanFlow, to practical large-scale generative models. We hope that MeanCache provides fresh insights for accelerating commercial-scale generative models.

ACKNOWLEDGMENTS

This work was supported by the National Natural Science Foundation of China Enterprise Innovation and Development Joint Fund Project U24B20181

ETHICS STATEMENT

This work does not introduce any new datasets or sensitive content, and all experiments are conducted on publicly available models. Furthermore, this study does not involve human subjects, animal experiments, or personally identifiable information. All datasets used are publicly available and strictly follow their usage licenses. We adhere to data usage guidelines throughout the experiments to ensure no ethical or privacy risks arise.

REPRODUCIBILITY STATEMENT

We have made every effort to ensure the reproducibility of our results. The experimental setup, including baseline parameter settings, model configurations, and hardware details, is described in detail in both the main paper and the supplementary material. In addition, we provide comprehensive ablation studies and analyses to further support reproducibility. To better illustrate the effectiveness of our approach, we also include high-resolution videos in the supplementary material. Furthermore, we plan to make key resources related to MeanCache, including core code (within permissible scope), available to the community. This is intended to encourage further exploration of MeanCache in compliance with relevant guidelines.

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

# MeanCache: From Instantaneous to Average Velocity for Accelerating Flow Matching Inference

# Appendix

We organize our appendix as follows:

**Proofs and Experimental Settings:**

**Additional Experimental Results and Analysis:**

**LLM Statement:**

## A  PROOFS AND EXPERIMENTAL SETTINGS

### A.1  MEANFLOW IDENTITY AT THE START POINT

**Start-point identity from the integral definition.**  Let $t < s$. In the MeanFlow formulation, the average velocity over the interval $[s, t]$ is defined as

$$u(z_s, t, s) = \frac{1}{s - t} \int_t^s v(z_\tau, \tau) \, d\tau. \tag{16}$$

Since the average velocity is uniquely determined by the interval $[s, t]$, it can be equivalently indexed by the state at the start point $z_t$. For notational consistency with the original MeanFlow identity, we write

$$u(z_t, t, s) \; = \; u(z_s, t, s). \tag{17}$$

Equivalently, the definition can be expressed as

$$(s - t) \, u(z_t, t, s) \; = \; \int_t^s v(z_\tau, \tau) \, d\tau. \tag{18}$$

**Differentiate both sides with respect to $t$.**

While the original MeanFlow identity is obtained by differentiating with respect to the end variable $s$, here we instead differentiate the definition equation 18 with respect to the start variable $t$ (holding $s$ fixed).

On the left-hand side, by the product rule,

$$\partial_t\big[(s - t) \, u(z_t, t, s)\big] \; = \; -u(z_t, t, s) \; + \; (s - t) \, \partial_t u(z_t, t, s). \tag{19}$$

On the right-hand side, by the Leibniz rule for a lower limit depending on $t$,

$$\partial_t\left(\int_t^s v(z_\tau, \tau) \, d\tau\right) \; = \; -v(z_t, t). \tag{20}$$

Equating equation 19 and equation 20 and rearranging yields the **MeanFlow Identity at the start point**:

$$v(z_t, t) \; = \; u(z_t, t, s) \; - \; (s - t) \frac{d}{dt} u(z_t, t, s). \tag{21}$$

## A.2 MODEL CONFIGURATION

For MeanCache, the primary hyperparameter is the peak-suppression parameter $\gamma$. Based on ablation studies, different $\gamma$ values are adopted across the three models to achieve better results. For the step size parameter $K$ of JVP caching, values from 2 to 5 are considered. Trajectory-Stability Scheduling is further applied in combination with multigraph construction and the Peak-Suppressed Shortest Path to determine the optimal acceleration path. Under different constraints $\mathcal{B}$, where $\mathcal{B}$ denotes the number of full computation steps, MeanCache flexibly balances runtime cost and acceleration ratio. The detailed parameter settings are summarized in Table 4.

Table 4: MeanCache configuration across different models.

| Model | Peak-suppression $\gamma$ | Budget $\mathcal{B}$ |
|---|---|---|
| FLUX.1[dev] | 4 | [10, 15] |
| Qwen-Image | 4 | [10, 13, 15] |
| HunyuanVideo | 3 | [10, 12] |

## A.3 BASELINES GENERATION SETTINGS

Experiments are conducted on three representative models from different tasks: FLUX.1-[dev] (Labs, 2024) and Qwen-Image (Wu et al., 2025) for text-to-image generation, and Hunyuan-Video (Kong et al., 2024) for text-to-video generation. The detailed configuration for each model is summarized below.

- **FLUX.1[dev]** (Labs, 2024): TeaCache (Liu et al., 2024) is evaluated with accumulation error thresholds $l = 0.25$ and $1.5$, corresponding to different acceleration ratios. DiCache (Bu et al., 2025) replaces the threshold discriminator of the first block in TeaCache with a probe-based perspective, where $\delta$ serves as the control factor. In the experiments, $\delta$ is set to 0.8 and 2.0. TaylorSeer (Liu et al., 2025) reformulates cache reuse as cache prediction, where $\mathcal{N}$ denotes the caching interval and $O$ the order of derivatives. For moderate acceleration (2.50–2.91×), the settings $\mathcal{N} = 6, O = 1$ and $\mathcal{N} = 6, O = 2$ are adopted, while a higher acceleration setting of $\mathcal{N} = 20, O = 1$ is applied to align with MeanCache.

- **Qwen-Image** (Wu et al., 2025): DBCache (vipshop.com, 2025) adjusts acceleration via $r$, with default hyperparameters $F_n = 2$ and $B_n = 4$. The compositional setting DB-Cache+TaylorSeer is further considered, configured with $r = 1.5$ and $O = 4$. Community implementations of TeaCache are also included, although significant quality degradation is observed when thresholds are large.

- **HunyuanVideo** (Kong et al., 2024): In addition to TeaCache, DiCache, and TaylorSeer, comparisons are conducted with ToCa (Zou et al., 2024a) and DuCa (Zou et al., 2024b), both configured with $\mathcal{N} = 5$, consistent with the TaylorSeer setting. Since TaylorSeer frequently encounters out-of-memory (OOM) under HunyuanVideo, all methods are evaluated with CPU-offload enabled to ensure fairness.

## A.4 METRICS

For fair and consistent evaluation, we consider both efficiency and generation quality. Efficiency is quantified using FLOPs and runtime latency. Quality is evaluated with the following five metrics:

**ImageReward.** ImageReward (Xu et al., 2023) is a learned metric designed to align with human preferences in text-to-image generation. It uses a reward model trained on human-annotated comparisons to provide scores that capture both fidelity and semantic alignment with the input text. Higher scores indicate better alignment with human judgment.

**CLIP Score.** CLIP Score (Hessel et al., 2021) leverages pretrained CLIP embeddings to evaluate text–image alignment. It computes the cosine similarity between image and text embeddings, with higher values indicating stronger semantic alignment. This metric complements ImageReward by

providing an embedding-based, zero-shot evaluation of alignment quality. For more precise assessment, we adopt ViT-G as the visual encoder in this paper.

**VBench.** VBench (Huang et al., 2024) evaluates video generation quality along 16 dimensions, including Subject Consistency, Background Consistency, Temporal Flickering, Motion Smoothness, Dynamic Degree, Aesthetic Quality, Imaging Quality, Object Class, Multiple Objects, Human Action, Color, Spatial Relationship, Scene, Appearance Style, Temporal Style, and Overall Consistency. We adopt the official implementation and weighted scoring to comprehensively assess video quality. In practice, we randomly select 350 generated video samples, which are evenly distributed across the 16 evaluation dimensions, and evaluate them using the official VBench protocol to ensure fairness and consistency.

**PSNR.** Peak Signal-to-Noise Ratio (PSNR) is widely used to measure pixel-level fidelity:

$$\text{PSNR} = 10 \cdot \log_{10}\left(\frac{R^2}{\text{MSE}}\right), \tag{22}$$

where $R$ is the maximum possible pixel value and MSE is the mean squared error between reference and generated images. Higher PSNR indicates better reconstruction fidelity. For videos, we compute PSNR per frame and report the average.

**LPIPS.** Learned Perceptual Image Patch Similarity (LPIPS) (Zhang et al., 2018) measures perceptual similarity using deep features:

$$\text{LPIPS} = \sum_i \alpha_i \cdot \text{Dist}(F_i(I_1), F_i(I_2)), \tag{23}$$

where $F_i$ denotes feature maps from a pretrained network, Dist is typically the $L_2$ distance, and $\alpha_i$ are layer-specific weights. Lower LPIPS values indicate higher perceptual similarity.

**SSIM.** The Structural Similarity Index Measure (SSIM) (Wang & Bovik, 2002) evaluates luminance, contrast, and structural consistency:

$$\text{SSIM}(x, y) = \frac{(2\mu_x\mu_y + C_1)(2\sigma_{xy} + C_2)}{(\mu_x^2 + C_1)(\mu_y^2 + C_1)(\sigma_x^2 + \sigma_y^2 + C_2)}, \tag{24}$$

where $\mu_x, \mu_y$ are mean values, $\sigma_x^2, \sigma_y^2$ are variances, $\sigma_{xy}$ is covariance, and $C_1, C_2$ are constants for stability. SSIM ranges from $-1$ to $1$, with larger values indicating stronger structural similarity.

## B  ADDITIONAL EXPERIMENTAL RESULTS AND ANALYSIS

### B.1  MEANCACHE VS. DISTILLATION

To better understand the difference between training-free caching and retraining-based distillation, we compare them on the commercial-scale text-to-image model Qwen-Image. Caching requires no extra training, whereas distillation depends on costly large-scale retraining. As Qwen-Image is newly released, distilled variants are limited; we therefore include the 15-step distilled model from DiffSynth-Studio as a representative baseline. Here, NFE (Number of Function Evaluations) denotes the number of sampling steps during inference. Table 5 shows that the distilled model achieves the highest ImageReward score, but MeanCache remains competitive, reaching 1.142 with only 10 steps. On CLIP Score, a measure of text–image alignment, MeanCache performs favorably. Distilled models also tend to drift from the original outputs, while caching better preserves fidelity to the backbone. This is reflected in reconstruction metrics, where MeanCache achieves lower LPIPS and higher SSIM and PSNR. Overall, distillation can improve some aspects through additional training, but our results suggest that caching-based approaches such as MeanCache provide a practical alternative. They maintain semantic consistency, deliver good perceptual quality, and achieve strong reconstruction accuracy, all without retraining. Caching thus appears to be a scalable acceleration strategy for commercial-grade generative models.

Table 5: Quantitative comparison between distillation and MeanCache on Qwen-Image.

| Method Qwen-Image | NFE | Training-Free | Visual Quality | | | | |
|---|---|---|---|---|---|---|---|
| | | | ImageReward ↑ | CLIP Score ↑ | LPIPS ↓ | SSIM ↑ | PSNR ↑ |
| Original (50 steps) | 50 | – | 1.180 | 33.626 | – | – | – |
| Qwen-Image-Distill-Full | 15 | ✗ | **1.162** | 33.062 | 0.594 | 0.505 | 11.019 |
| Qwen-Image-Distill-Full | 10 | ✗ | 1.118 | 32.802 | 0.583 | 0.524 | 11.483 |
| MeanCache | 15 | ✓ | 1.159 | **33.636** | **0.075** | **0.938** | **27.663** |
| MeanCache | 10 | ✓ | 1.142 | 33.621 | 0.236 | 0.815 | 18.983 |

## B.2 EFFICIENCY–PERFORMANCE TRADE-OFF

Our analysis in Fig. 8 compares MeanCache, TaylorSeer, TeaCache, and DiCache on FLUX.1[dev] across three reconstruction metrics (LPIPS, SSIM, and PSNR). Across a wide range of latency configurations, MeanCache consistently delivers higher reconstruction fidelity while requiring less inference time. This advantage is particularly clear in the low-latency regime: even when the total runtime falls below 3 seconds, MeanCache maintains stable performance across all metrics, whereas baseline methods exhibit severe degradation, including loss of structural consistency and perceptual quality. These results highlight that MeanCache not only achieves a favorable quality–efficiency balance, but also extends the usable acceleration range far beyond prior caching strategies, enabling practical deployment in interactive or resource-constrained scenarios.

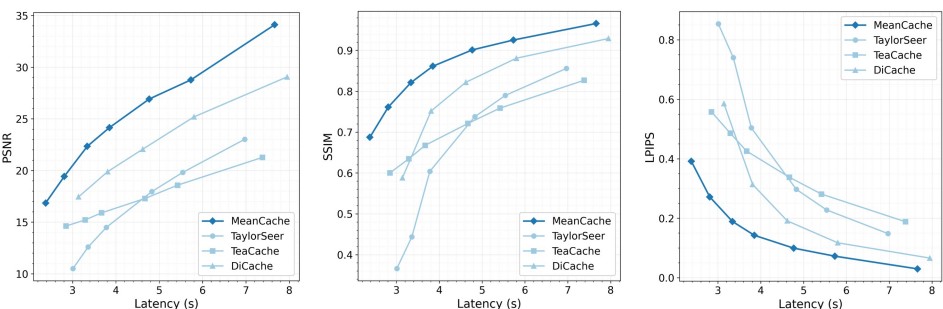

Figure 8: Efficiency–Performance Trade-off on FLUX.1[dev].

## B.3 ADDITIONAL QUALITATIVE RESULTS

We provide further qualitative results to complement the main paper. These experiments cover high-acceleration scenarios on Qwen-Image, different acceleration ratios on FLUX.1[dev], rare-word prompts for testing content consistency, and additional video examples on HunyuanVideo.

**Qwen-Image under High Acceleration.** Figure 9 shows results on Qwen-Image at high acceleration ratios. Compared with baselines, MeanCache achieves more faithful preservation of structural details and visual quality, even when runtime is significantly reduced. Competing methods display blurring or noticeable artifacts, while MeanCache remains robust.

**FLUX.1 across Acceleration Ratios.** Figures 10 and 11 illustrate results on FLUX.1[dev] across multiple acceleration ratios. MeanCache consistently outperforms TaylorSeer and TeaCache in preserving fidelity and perceptual quality. Notably, even at a $4.12\times$ speedup, where baselines collapse in quality, MeanCache maintains coherent textures and stable global structure.

**Content Consistency under Rare-Word Prompts.** In Figure 12, we compare MeanCache with TaylorSeer and TeaCache using the rare-word prompt *"Peristeronic"*. While all methods retain reasonable content under mild acceleration ($< 2.5\times$), TaylorSeer and TeaCache quickly degrade as speedup increases, showing severe semantic drift and detail loss. In contrast, MeanCache preserves both the intended concept and fine-grained details even at $4.03\times$ acceleration.

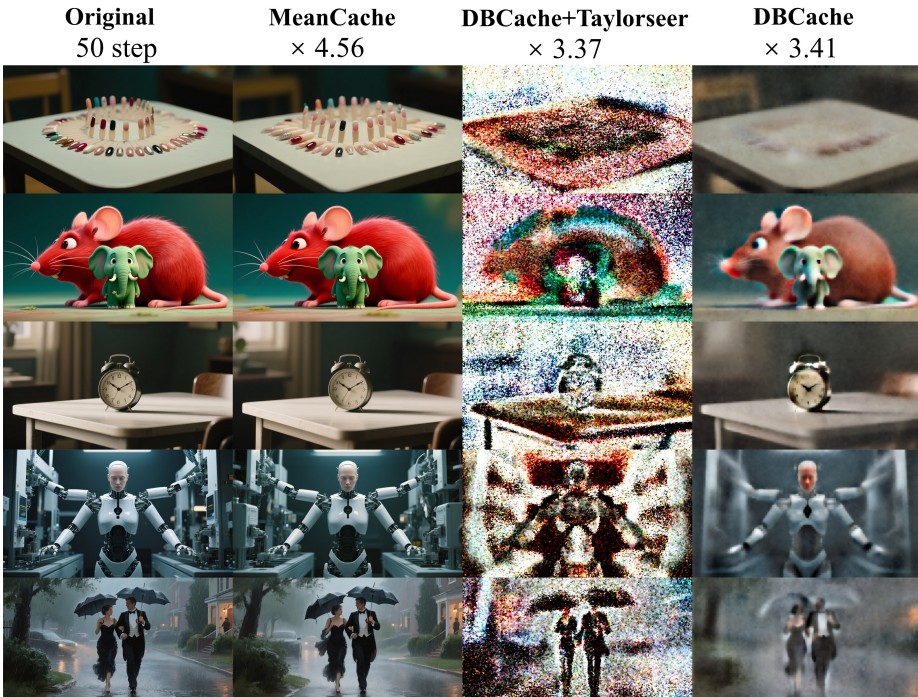

Figure 9: Qualitative comparison of different methods at high acceleration ratios on **Qwen-Image**.

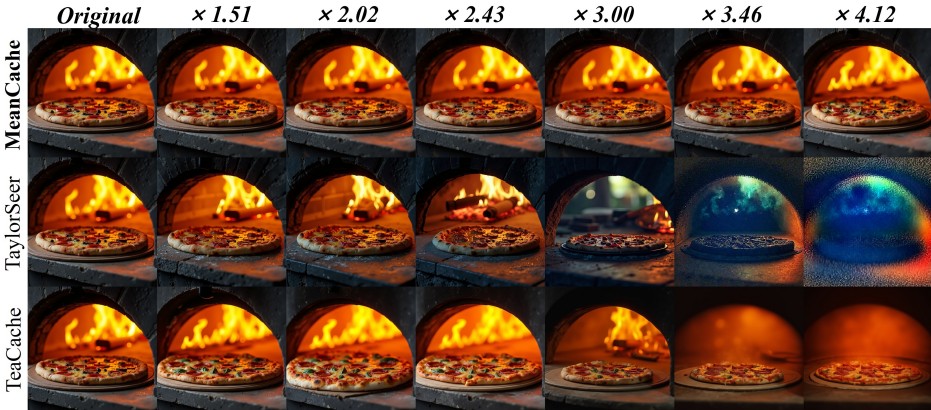

Figure 10: Qualitative comparison on **FLUX.1[dev]** under different acceleration ratios (Supplementary 1).

**HunyuanVideo under High Acceleration.** Figure 13 presents supplementary video results on HunyuanVideo. When the acceleration exceeds $3\times$, baseline methods suffer from heavy motion artifacts, visual degradation, and temporal inconsistency. MeanCache, however, continues to deliver stable frame quality and temporal coherence, closely matching the reference trajectory and confirming its robustness in video generation tasks.

## C USE OF LARGE LANGUAGE MODELS

We used large language models (LLMs) solely for grammar checking and minor language polishing. No part of the method design, experimental setup, analysis, or results was generated by LLMs. All technical contributions and empirical findings in this paper are entirely by the authors.

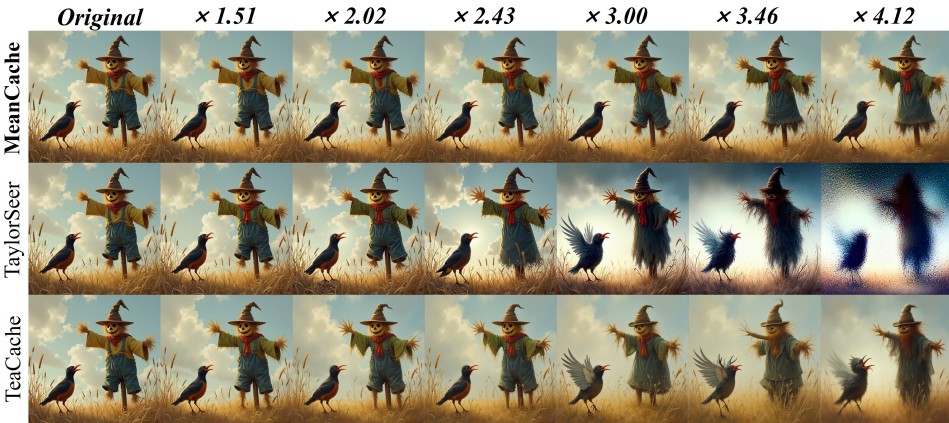

Figure 11: Qualitative comparison on **FLUX.1[dev]** under different acceleration ratios (Supplementary 2).

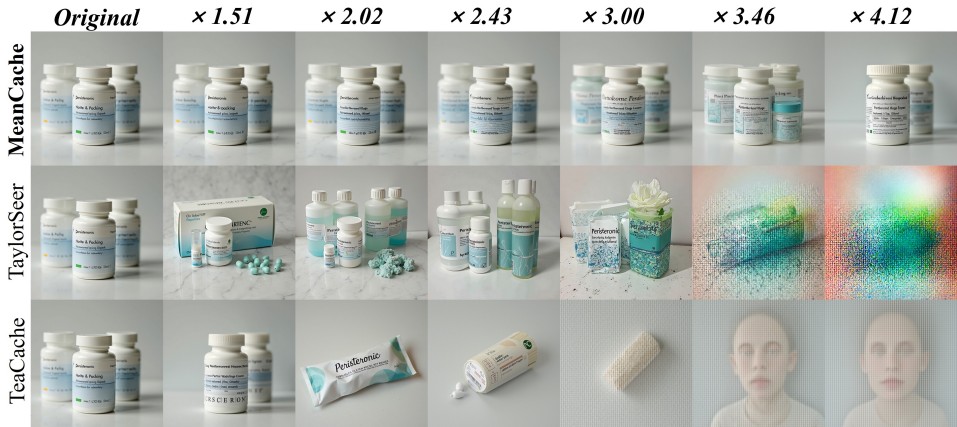

Figure 12: Content consistency under rare-word prompts "Peristeronic" across acceleration ratios.

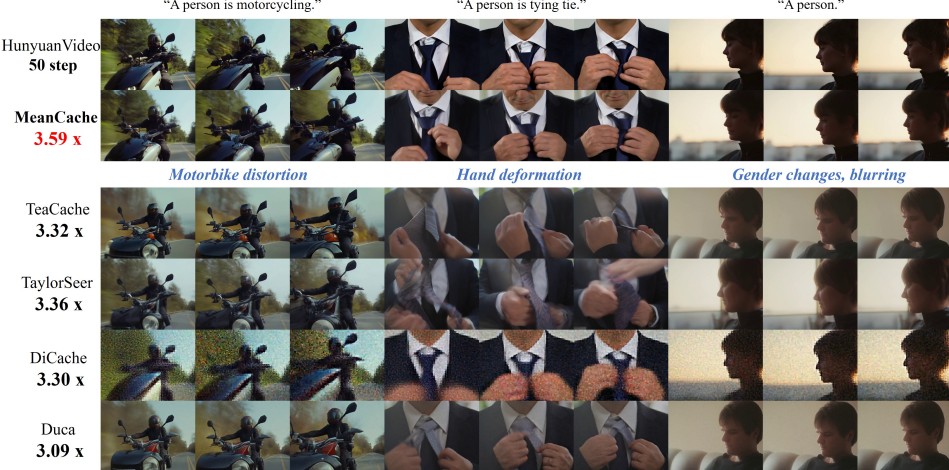

Figure 13: Comparison of different methods at high acceleration ratios on **HunyuanVideo**.

