# OpenReview forum: "MeanCache: From Instantaneous to Average Velocity for Accelerating Flow Matching Inference"
_ICLR.cc/2026/Conference — ICLR 2026 Poster_

### Official Review · Reviewer_pV1g · 2025-10-23

**Soundness:** 1
**Presentation:** 2
**Contribution:** 2
**Rating:** 2
**Confidence:** 4

**Summary:**

The paper introduces MeanCache, a training-free caching framework designed to accelerate flow model inference in large-scale generative models. Traditional caching methods reuse intermediate representations to save computation but rely on instantaneous velocity fields, which lead to error accumulation. MeanCache addresses this by reframing caching in the average-velocity domain, inspired by the MeanFlow formulation.

**Strengths:**

Overall, the paper’s motivation and structure are sound. I was able to follow the authors’ intuitions and main ideas, though the mathematical formulations lack sufficient explanation and rigor (see weaknesses). Additionally, I appreciate the comprehensive empirical evaluation on large-scale generative models.

**Weaknesses:**

The biggest concern I have is the claim that MeanFlow-based caching is more robust than caching in Flow Matching. Although the authors provide an explanatory figure (Fig. 3), this is not convincing, since error accumulation can also be observed in MeanFlow. Can the authors provide high-level intuition and mathematical proofs to justify this claim? Additionally, MeanFlow is an existing idea. The main contributions of the paper come from implementing feature caching within the MeanFlow model. To assess the core contribution, I would like to see how vanilla MeanFlow (without caching) performs. This would help isolate and evaluate the effects of the proposed method more clearly.


I find some mathematical formulations and claims inaccurate and presented without sufficient explanation.
First, the JVP approximation error can be large when $t - r \gg \delta$. Can the authors show whether this error is negligible (or quantify it)?
Second, the metric used in Eq. (13) depends on timestep discretization, which can vary across applications and the number of sampling steps. This is also connected to my first point, since $u(z_s, t, s)$ is already an approximation of the true vector in Eq. (2). However, I do not find discussions or mathematical proofs that show the approximation of the error.
Lastly, I found some minor details regarding the formulations:


- L.106 lower case paragraph not in consistency
- Eq. (1) $\hat{x} $ is introduced without explanation
- Eq. (2) is $s > t$? This indicates forward velocity towards prior? Might be better to represent in reverse time consistently in line with L.122 $x_1 - x_0$ sounds more natural.
- Eq. (4) $\mathcal{F}$ and l are not defined elsewhere
- Eq. (5) JVP is a function of $z, t, s$. Might be better to express explicitly instead of $\hat{JVP}$
- Eq. (7) $\hat{JVP} \approx \frac{d}{dr}$. Not equality by definition.
- Eq. (15) no explanation of $\mathcal{B}$

**Questions:**

Why caching in MeanFlow would be better than caching in flow models? JVP term is approximated in MeanFlow too. This part is most confusing and concerning to me.

---

> ### Author Response · Authors · 2025-11-24
> **Response to Reviewer  pV1g (1/3)**
>
> > **W1 & Q1: The main contributions of the paper come from implementing feature caching within the MeanFlow model. Why would caching in MeanFlow be better than caching in flow models?**
>
> Thank you for the question. We would like to clarify an important point:
>
> | **MeanCache is not a ‘feature caching method’ within the ‘MeanFlow’ model**.
>
> Instead, we just inspired the insight (average velocity) from MeanFlow and apply it to the caching problem for all FM models. Let us explain in detail:
>
> **(1) No MeanFlow-trained commercial models are available.**\
> Current commercial generative models (e.g., FLUX, Qwen-Image, HunyuanVideo) do not provide any usable MeanFlow-trained versions. Therefore, there is no “MeanFlow model” at this commercial-model scale that can be directly used for a fair caching comparison.
>
>
> **(2) MeanFlow and MeanCache serve fundamentally different purposes.**
>
> - **MeanFlow** is a training method aimed at improving generative modeling by incorporating mean velocity during training, without relying on distillation or curriculum learning. While it performs well on smaller datasets such as CIFAR-10 and ImageNet-256, applying MeanFlow training to commercial-scale models remains challenging.
> - **MeanCache** is a training-free, plug-and-play acceleration method that can be directly applied to existing commercial models. Although the insight on mean velocity was inspired by MeanFlow, we re-derive the formulation specifically for caching. In our method, JVP caching together with the Stability Map jointly enhances acceleration quality, achieving significantly better stability and lower error accumulation than existing caching approaches.
>
> **(3) JVP Caching is Not Feature Caching**\
> Existing feature caching methods (e.g., TeaCache, TaylorSeer) focus on predicting instantaneous velocity. In contrast, JVP caching aims to predict the mean velocity, not the instantaneous velocity. The theoretical foundation of MeanCache is based on the MeanFlow identity at the start point (Eq. 21), where the concept of average velocity is used to provide more stable and robust predictions.
>
> In summary, our approach does not perform caching within the MeanFlow model itself. Instead, we apply the conceptual insights from MeanFlow to design a training-free, deployment-ready acceleration method tailored for commercial-scale generative models.
>
> ---
>
> > **W2: Theoretical Intuition: Why MeanCache is More Robust:**
>
> We attribute the robustness of MeanCache to three fundamental mathematical advantages:
>
> **(1) Smoother Target Manifold (Integral vs. Instantaneous).**
> Standard methods approximate instantaneous velocity $v_t$, which acts as the derivative of the trajectory. In complex generation, $v_t$ fluctuates rapidly (high curvature, large $\|\nabla^2 v\|$), making extrapolation error-prone.
> In contrast, MeanCache approximates average velocity $u_{[t,s]} = \frac{1}{s-t}\int_t^s v_\tau d\tau$. Mathematically, integration acts as a low-pass filter, smoothing out high-frequency oscillations. Since the manifold of average velocity is significantly smoother than that of instantaneous velocity ($\|\nabla^2 u\| \ll \|\nabla^2 v\|$), the JVP approximation inherently yields lower truncation errors.
>
> **(2) Stability-Aware Scheduling (The Stability Map).**
> In contrast to baselines that rely on greedy heuristics (e.g., TeaCache) or rigid fixed intervals (e.g., TaylorSeer) for uncertainty prediction, MeanCache is guided by a pre-computed Stability Map.
> By modeling the error distribution as a stability graph, we explicitly localize regions of rapid fluctuations. Coupled with the Peak-Suppressed mechanism, our approach jointly optimizes both the caching timesteps and the optimal reference span $K$ (the interval $r \rightarrow t$). This ensures that JVP caching is activated only within verified stable intervals using the most appropriate $K$, while enforcing exact full computation in unstable regions to rigorously mitigate risk.
>
> **(3) Error Bound.**
> Eq. (11) defines the MeanCache estimator via the cache span $K$.
> For $K = 1$, the JVP term vanishes and **MeanCache reduces exactly to instantaneous-velocity caching**, which serves as an explicit upper bound on MeanCache’s approximation error. Therefore, MeanCache’s worst-case error on any interval $t \to s$ never exceeds that of instantaneous caching. A more detailed discussion of this theoretical guarantee is provided in our responses to **Reviewer 4J2V W1** and **Reviewer 5MTc W1**.

---

> ### Author Response · Authors · 2025-11-25
> **Response to Reviewer pV1g (2/3)**
>
> ---
> > **W3: We would like to see the performance of vanilla MeanFlow without caching, to separate the effects of MeanFlow itself from the proposed method.**
>
> Thank you for the insightful question. As mentioned in Q1, there are currently no commercially available generative models with a fully trained MeanFlow version. Therefore, we cannot conduct a direct “separation experiment” comparing the performance of a “vanilla MeanFlow” with our proposed method.
>
> **Indirect Experiment: Comparison on a Distilled Model**\
> However, we believe testing whether MeanCache can be applied to a **MeanFlow model** is meaningful. We identified a distilled model, **HunyuanImage-2.1-Distilled**, which incorporates ideas from MeanFlow and distills a model with fewer steps (8 steps) from **HunyuanImage-2.1 [1]**. (Note: this is not a true MeanFlow model.) We conducted a comparison on this distilled model to test the effectiveness of MeanCache.
>
> | Method                                     | Latency (s)↓ | Image Reward↑ | CLIP Score↑ | LPIPS↓ | SSIM↑ | PSNR↑  |
> | ------------------------------------------ | ------------ | ------------- | ----------- | ------ | ----- | ------ |
> | HunyuanImage-2.1                           | 26.59        | 1.103         | 32.605      | -      | -     | -      |
> | **HunyuanImage-2.1 + MeanCache**           | 8.73         | 1.082         | 32.642      | 0.143  | 0.898 | 24.444 |
> | HunyuanImage-2.1-Distilled                 | 2.73         | 1.050         | 31.780      | 0.505  | 0.688 | 12.583 |
> | **HunyuanImage-2.1-Distilled + MeanCache** | 2.19         | 1.043         | 31.893      | 0.502  | 0.692 | 12.547 |
>
>
> This demonstrates that MeanCache not only accelerates the original FM model (HunyuanImage-2.1) but also shows strong potential when applied to a distilled model (HunyuanImage-2.1-Distilled) inspired by MeanFlow.
>
> ---
>
> > **W4: When the time interval is large, the JVP approximation error may be significant. Can the authors demonstrate whether this error can be neglected or quantify the error?**
>
> Thank you for the question. As you pointed out, the JVP approximation error increases when the time interval is large. This error is not negligible; in fact, it is the core of our caching scheduling strategy (Section 2.4).
>
> In the preprocessing scheduling process, MeanCache quantifies errors from two sources: different values of $K$ and cache intervals. It prioritizes configurations with smaller errors for cache activation and reuse. We use the Peak-Suppressed Shortest Path algorithm to ensure the stability of the overall trajectory.
>
> ---

---

> ### Author Response · Authors · 2025-11-25
> **Response to Reviewer pV1g (3/3)**
>
> ---
>
> > **W5: The metric used in Eq. (13) depends on the timestep discretization, and different applications or sampling steps may lead to different discretizations.**
>
> Thank you for the insightful comment. Indeed, the metric in Eq. (13) depends on the timestep discretization, but we still believe that MeanCache remains an effective acceleration method. First, MeanCache can be customized for different timesteps or application scenarios, performing offline calculations to provide stable paths for the corresponding timesteps. Second, for most users, the model’s default timestep (provided by the official implementation) generally does not change frequently. Third, we believe this is a common issue for caching methods. All existing caching methods require a fixed timestep setting, which is paired with a corresponding strategy. When the timestep changes, hyperparameters or prior settings (such as thresholds) must be redefined.
>
> ---
>
> > **W6: Issues with the details of the formulas**
>
> Thank you very much for the reviewer’s feedback. It will greatly help improve the quality of our paper. We have updated the PDF according to your suggestions:
>
> - The lowercase paragraph inconsistency has been fixed (L.106).
> - The definition of $\hat{x}$ has been added (L.126).
> - The direction of the timestep has been revised (L.128, L.839).
> - The definitions for $\mathcal{F}$ and $I$ have been added (L.147).
> - The explanation of $\mathcal{B}$ has been supplemented (L.254).
> - Regarding the JVP symbol issue, we will consider a better expression and present it in the final PDF.
>
> Please let us know if additional clarification would be helpful. Once again, thank you for your valuable suggestions.
>
> ---
>
> **Reference**
>
> [1] Tencent Hunyuan Team. Hunyuanimage 2.1: An efficient diffusion model for high-resolution (2k) textto-image generation.  (2025)

---

### Official Review · Reviewer_5MTc · 2025-11-01

**Soundness:** 3
**Presentation:** 3
**Contribution:** 2
**Rating:** 6
**Confidence:** 4

**Summary:**

This paper introduces MeanCache, a training free caching framework for Flow Matching (FM) inference. The method reframes caching from the instantaneous velocity domain to the average velocity domain, using a JVP based estimator to approximate interval average velocities and mitigate local error accumulation. A trajectory stability scheduling strategy casts timestep selection as a peak suppressed shortest path problem on a multigraph; dynamic programming yields the cache placement under a step budget. Experiments on FLUX.1, Qwen Image, and HunyuanVideo report ∼3–4.5× speedups with competitive perceptual quality vs. strong caching baselines.

**Strengths:**

+ Clear, practical idea: Average velocity caching with a JVP estimator is simple yet impactful; it addresses the well known error accumulation issue at high acceleration ratios.
+ Scheduling formalization: Casting cache placement as a constrained shortest path problem is principled and provides a tunable trade off via $\gamma$ and budget $B$.
+ Strong empirical results: Consistent improvements over strong baselines across image and video; qualitative figures align with quantitative gains.

**Weaknesses:**

- Heuristic JVP reuse: Approximating $JVP_{t\to s}$ by $JVP_{r\to t}$ is empirically motivated. The paper does not provide theoretical guarantees or error bounds beyond the MeanFlow identities. Performance depends on the span $K$, which is tuned and schedule dependent.
- Stability assumption: The scheduling assumes that “relative changes at fixed timesteps are highly consistent across samples.” This is plausible but not statistically substantiated. Some quantification (e.g., cross prompt correlation of stability costs) would strengthen the claim.
- Cost of graph construction: The paper states the graph is built offline with 50 prompts $\times$ 5 seeds, but the actual overhead (FLOPs/GPU hours) and sensitivity to the prompt set are not reported. It remains unclear how often this needs to be rebuilt across domains or resolutions.

**Questions:**

- Schedule generalization: If the stability graph is built on one prompt pool/domain, how well does it transfer to another (e.g., different prompt sets or resolutions)? Any results on cross domain reuse?
- Offline cost: Please report the wall clock overhead (and FLOPs) to build the multigraph and the proportion relative to the downstream inference savings.

---

> ### Author Response · Authors · 2025-11-24
> **Response to Reviewer 5MTc (1/2)**
>
> > **W1: JVP reuse is heuristic and lacks formal theoretical guarantees.**
>
> We appreciate the reviewer’s comment. To address this concern, we provide three intuitive reasons for the stability of our approach, together with an error bound analysis, as detailed below:
>
> 1.  **Averaged velocities (MeanFlow) are inherently smoother than instantaneous velocities.**\
>     The averaged velocity reflects an integrated trend rather than pointwise fluctuations, leading to lower curvature and better extrapolation even in locally nonlinear regions.
>
> 2.  **A JVP encodes relational derivatives rather than raw dynamical values.**\
>     What we cache is not a feature or instantaneous velocity, but a derivative-based relation (Jacobian–vector product). Such relations tend to be more structured and stable, making local approximations more reliable than feature/velocity caching used in prior FM methods.
>
> 3.  **The scheduling mechanism avoids regions with strong nonlinearity and large approximation errors.**\
>     The cache scheduler selects an appropriate span (K) and uses the Peak-Suppressed Shortest Path strategy to choose the most stable reuse path, effectively preventing error accumulation. This stability-driven scheduling is not present in prior approaches such as TeaCache or TaylorSeer.
>
> **Error Bound Guarantee**
>
> A similar discussion is also provided in our response to Reviewer 4J2V’s comment W1.
>
> Eq. 11 defines the MeanCache estimator through the cache span $K$, i.e., the number of discrete steps over which the cached JVP is reused:
> $$
> \\widehat{u}(z_t,t,s)=
> \\begin{cases}
> v(z_t,t) + (s-t) \\widehat{\\mathrm{JVP}}_K, & K > 1, \\\\[6pt]
> v(z_t,t), & K = 1.
> \\end{cases}
> $$
>
> When \\(K>1\\), MeanCache leverages JVP caching to compensate for local approximation errors accumulated across the cache span. In contrast, when \\(K=1\\), the JVP term vanishes and **MeanCache reduces to instantaneous-velocity caching**, which represents the theoretical worst-case (upper-bound) error behavior. Since the Stability Map jointly optimizes over and can always fall back to either regime, the resulting MeanCache error is guaranteed not to exceed that of instantaneous caching. Therefore, we obtain:
> \\[
> E_{\text{MeanCache}} \le E_{\text{Inst}} .
> \\]
>
> Thus, MeanCache preserves a guaranteed error bound on any interval \\( t \\to s \\), ensuring that its error remains no worse than instantaneous caching.
>
> ---
> > **W2 & Q1: The claim of “cross-sample stability” lacks empirical support. If the stability map is built on one prompt pool/domain, how well does it transfer to another (e.g., different prompt sets or resolutions)?**
>
> Thank you for the insightful comment. We agree that additional evidence is important for demonstrating generality. To address this, we conducted two transfer experiments—cross-domain and cross-resolution—both on text-to-image generation, while reusing exactly the same stability map as in the paper. This stability map was constructed once from an independent text-to-video dataset and was applied directly without any recomputation or task-specific adaptation.
>
> **Cross-domain transfer.**\
> We directly reused the stability map on three challenging text-to-image sub-tasks. Despite the domain shift, MeanCache consistently outperforms TaylorSeer, even on out-of-domain prompt sets. Please refer to Reviewer Egs2 Q1-1 for detailed results.
>
> **Cross-resolution transfer.**\
> We further applied the stability map generated at **1024×1024** to **1024×576 (16:9)** without modification. As shown below, MeanCache retains clear advantages across all visual-quality metrics, demonstrating strong robustness to resolution changes.
>
> | Method               | Resolution | Image Reward↑ | CLIP Score↑ | LPIPS↓ | SSIM↑ | PSNR↑  |
> | -------------------- | ---------- | ------------- | ----------- | ------ | ----- | ------ |
> | Original             | 1024×1024  | 1.033         | 31.229      | –      | –     | –      |
> | TaylorSeer (N=6,O=1) | 1024×1024  | 0.961         | 31.191      | 0.419  | 0.660 | 15.831 |
> | **MeanCache (B=15)** | 1024×1024  | **1.010**         | **31.244**      | **0.142**  | **0.870** | **24.834** |
> | Original             | 1024×576   | 0.728         | 30.153      | –      | –     | –      |
> | TaylorSeer (N=6,O=1) | 1024×576   | 0.708         | 29.093      | 0.398  | 0.652 | 16.401 |
> | **MeanCache (B=15)** | 1024×576   | **0.711**         | **30.153**      | **0.105**  | **0.891** | **26.631** |

---

> ### Author Response · Authors · 2025-11-24
> **Response to Reviewer 5MTc (2/2)**
>
> ---
> > **W3 & Q3 Offline cost and the proportion relative to the downstream inference savings**
>
>
> Thank you for the question. The dominant offline overhead comes from constructing the Stability Map, whose total wall-clock cost in our experiments is 6.7–16.8 minutes. The Peak-Suppressed Shortest-Path scheduling itself is negligible (<0.02s). More detailed statistics can be found in Reviewer Egs2 W3 & Q4.
>
>
> In practice, the Stability Map is highly insensitive to seeds, and using 50 prompts is overly conservative; around 20 prompts are typically sufficient. To quantify the cost relative to downstream inference savings, we compute the break-even points, i.e., the number of generated images after which the offline preprocessing time is fully amortized. Using FLUX.1 as an example, the per-image saving under MeanCache (B=10) is:
>
> Δ = 11.57 − 2.81 = 8.76s per image
>
> With this saving, the offline construction cost is amortized after:
>
> *   **50 prompts:** 50 × 20.0 / 8.76 ≈ 115 images
> *   **20 prompts:** 20 × 20.0 / 8.76 ≈ 46 images
>
> These results show that the one-time offline cost is quickly amortized by the inference savings(especially in real-world online deployment scenarios). Moreover, Stability Map construction is fully parallelizable across prompts; multi-GPU execution can greatly reduce the wall-clock time.

---

### Official Review · Reviewer_4J2V · 2025-11-01

**Soundness:** 3
**Presentation:** 4
**Contribution:** 3
**Rating:** 6
**Confidence:** 4

**Summary:**

This paper presents MeanCache, a training-free framework to accelerate Flow Matching inference. It addresses error accumulation in prior caching methods by shifting from an instantaneous to an average velocity perspective. The core idea is to approximate the interval average velocity using a cached Jacobian-Vector Product for better stability. This is combined with a principled scheduling strategy, formulated as a shortest-path problem. Experiments on large-scale models (FLUX, Qwen-Image, HunyuanVideo) demonstrate significant speedups while maintaining or improving generation quality over state-of-the-art caching baselines.

**Strengths:**

1. MeanCache consistently outperforms state-of-the-art baselines, especially at high acceleration ratios where competing methods collapse.
2. The inclusion of both perceptual metrics and reconstruction metrics provides thorough quality assessment.
3. The paper is very well-written, clearly motivating the problem and lucidly explaining the proposed methodology.

**Weaknesses:**

1. The paper lacks theoretical analysis (e.g., error bounds) explaining why JVP-based average velocity outperforms TaylorSeer's Taylor expansion, leaving the source of empirical gains unclear.
2. Constructing the multigraph and computing shortest paths incurs preprocessing cost. Table 1-2 report only inference latency, not total time including preprocessing.
3. TaylorSeer encounters OOM on HunyuanVideo, forcing CPU-offload for all methods. This may artificially inflate latency measurements and doesn't reflect realistic deployment scenarios.

**Questions:**

1. How does the preprocessing time scale with budget $B$?
2. Is the stability map computed on 50 prompts transferable to the entire prompt distribution?
3. What is the memory footprint of storing cached JVPs and the multigraph?

---

> ### Author Response · Authors · 2025-11-24
> **Response to Reviewer 4J2V (1/2)**
>
> > **W1: The paper lacks theoretical analysis explaining why JVP-based average velocity outperforms TaylorSeer's Taylor expansion, leaving the source of empirical gains unclear.**
>
> Thank you for the comment. We believe the empirical performance of MeanCache is superior to TaylorSeer for the following reasons:
>
> 1.  **Instantaneous velocity vs. average velocity:**\
>     TaylorSeer relies on feature caching, which aims to correct instantaneous velocity. However, instantaneous velocities are inherently fluctuating (as shown in Fig. 2), whereas average velocities are smoother and more linear. This makes average velocity easier to predict and more stable, which naturally leads to improved performance in the context of cache reuse.
>
> 2. **Stability Map: adaptive scheduling of cache timing and JVP span $K$**
>     TaylorSeer uses a fixed cache interval and a fixed Taylor expansion order, which cannot adapt when the flow becomes non-linear. In contrast, MeanCache builds a stability map that evaluates the local approximation deviation:
>
>     $$
>     \mathcal{L}_K(t,s)=\|u(z_t,t,s)-\hat{u}(z_t,t,s)\|.
>     $$
>
>     By selecting the cache configuration with the lowest predicted deviation(Eq. 15), MeanCache automatically determines
>     - **when** to reuse cached information, and
>     - **how far** to reuse a JVP (the cache span $K$).
>
>     This adaptive scheduling prevents error spikes and avoids unstable regions—something fixed-interval Taylor expansion cannot do.
>
> **Theoretical Analysis: Error Bound Guarantee**
>
> Eq. 11 defines the MeanCache estimator through the cache span $K$, i.e., the number of discrete steps over which the cached JVP is reused:
> $$
> \\widehat{u}(z_t,t,s)=
> \\begin{cases}
> v(z_t,t) + (s-t) \\widehat{\\mathrm{JVP}}_K, & K > 1, \\\\[6pt]
> v(z_t,t), & K = 1.
> \\end{cases}
> $$
>
> When \\(K>1\\), MeanCache leverages JVP caching to compensate for local approximation errors accumulated. In contrast, when \\(K=1\\), the JVP term vanishes and **MeanCache reduces to instantaneous-velocity caching**, which represents the theoretical worst-case (upper-bound) error behavior. Since the Stability Map jointly optimizes over and can always fall back to either regime, the resulting MeanCache error is guaranteed not to exceed that of instantaneous caching. Therefore, we obtain:
> \\[
> E_{\text{MeanCache}} \le E_{\text{Inst}} .
> \\]
>
> Thus, MeanCache preserves a guaranteed error bound on any interval \\( t \\to s \\), ensuring that its error remains no worse than instantaneous caching.
>
>
>
> ---
>
>
> > **W2: Preprocessing Cost Analysis**
>
> Thank you for the question. The preprocessing consists of two main parts, with most of the cost coming from constructing the Stability Map. This involves running the original denoising trajectory, which depends on the model’s inference time, and computing the error between the true and predicted average velocity, which consistently takes about 8.15 seconds per sample.
>
> In comparison, the Peak-Suppressed Shortest-Path Scheduling introduces almost no additional overhead.
>
> In our experiments (20-50 samples), constructing the Stability Map requires 6.7-16.8 minutes in total, whereas the shortest-path computation is negligible (<0.02s). A more detailed breakdown is provided in Reviewer Egs2’s W3 & Q4.
>
> ---
>
> > **W3: CPU-offload for all methods, which may not reflect realistic deployment scenarios.**
>
> Thank you for the question. On HunyuanVideo, TaylorSeer’s memory usage exceeds the capacity of an H100/80GB GPU, resulting in OOM. To ensure fairness, we applied CPU-offload uniformly to all methods so that performance is not confounded by hardware differences.
>
> In practical deployment, MeanCache itself does not require CPU-offload on H100/80GB. The table below reports MeanCache’s runtime with and without CPU-offload, along with the relative improvement:
>
> | **MeanCache** | **w**  | **w/o** | **Time Reduction (%)** |
> | ------------- | ------ | ------- | ---------------------- |
> | **B = 12**    | 33.05s | 28.69s  | 13.19%                 |
> | **B = 10**    | 29.48s | 25.23s  | 14.42%                 |
>
> As shown, MeanCache reduces time by 13.19% and 14.42% without CPU-offload, improving inference efficiency compared to the offload setting.

---

> > ### Author Response · Authors · 2025-11-24
> > **Response to Reviewer 4J2V (2/2)**
> >
> > ---
> >
> > > **Q1: How does preprocessing time vary with budget $B$?**
> >
> > Thank you for the question. The $B$ primarily affects the Peak-Suppressed Shortest-Path Scheduling step. As explained in **W2**, this step has minimal time cost(millisecond-level) after the Stability Map is built. However, there are slight variations in cost for different values of $B$, as shown in the table below on FLUX.1 with H100/80GB:
> >
> > | **$B$** | **Cost (sec)** |
> > | ------------ | -------------- |
> > | 10           | 0.004          |
> > | 15           | 0.008          |
> > | 20           | 0.012          |
> > | 30           | 0.018          |
> > | 40           | 0.022          |
> > ---
> >
> > > **Q2: Is the stability map computed on 50 prompts transferable to the entire prompt distribution?**
> >
> > Thank you for the question. Yes, the stability map built from 50 prompts can be generalized to the entire prompt distribution.
> >
> > Take FLUX.1 as an example, our stability map is constructed using an independent text-to-video dataset, which is typically Out-Of-Domain(OOD) setting for FLUX.1. Despite the distribution gap, we observed strong generalization when evaluating on text-to-image benchmarks.
> >
> > To further verify this, we reused the same cache schedule on three challenging sub-tasks without recomputing the stability map, and MeanCache consistently outperformed TaylorSeer, even for out-of-distribution cases.
> >
> > For more details, please refer to **Reviewer Egs2 Q1-1** and **Reviewer 5MTc W2 & Q1**.
> >
> > ---
> >
> > > **Q3: What is the memory footprint of storing cached JVPs and the multigraph?**
> >
> > Thank you for the question.
> > During multigraph construction (offline), we run one full trajectory, record the required instantaneous velocities, latent states, and timesteps, and compute the stability scores (Eqs. 13–14). This recording and computation can be performed entirely on the CPU, and the multigraph is a lightweight CPU-side structure with essentially no GPU memory usage. Therefore, GPU memory consumption in this stage is mainly due to running the original trajectory itself.
> >
> > During inference (online), MeanCache uses JVP caching, similar to other cache-based methods, which stores a few latent tensors and therefore introduces only a small additional GPU memory cost.
> >
> > We measured GPU memory on FLUX.1 using torch.cuda.max\_memory\_allocated() under three settings:
> >
> > | Setting                     | Mode               | Peak GPU Memory (GB) |
> > | --------------------------- | ------------------ | -------------------- |
> > | No Cache                    | Online inference   | 38.37                |
> > | **JVP Cache**               | Online inference   | **38.77 (+0.40)**    |
> > | **Multigraph Construction** | Offline  | **36.35**            |
> >
> > As shown, both multigraph construction and JVP caching remain close to the original inference memory footprint. JVP caching increases peak GPU memory by only 0.40 GB (\~1%)

---

### Official Review · Reviewer_Egs2 · 2025-11-01

**Soundness:** 4
**Presentation:** 3
**Contribution:** 3
**Rating:** 6
**Confidence:** 3

**Summary:**

This paper proposes MeanCache, a training-free caching framework for accelerating Flow Matching diffusion models. Unlike prior methods that cache instantaneous model outputs (velocities) at fixed intervals, MeanCache operates in the average-velocity domain. Specifically, it uses cached Jacobian–vector products (JVPs) from earlier timesteps to construct interval-average velocities over longer spans, which are significantly smoother than instantaneous velocities. By reusing these interval-average velocities, MeanCache mitigates the local error accumulation that plagues existing caching schemes. To decide when and how far to cache, the paper introduces a trajectory-stability scheduling strategy: denoising steps are nodes in a graph, edges represent caching intervals weighted by their induced velocity-error, and a peak-suppressed shortest-path search (a budget-constrained graph optimization) selects the caching schedule that avoids large error spikes. In experiments on large-scale models (FLUX.1 image model, Qwen-Image, and HunyuanVideo), MeanCache achieves substantial speedups (e.g. 4.12×, 4.56×, and 3.59×) while maintaining or even improving generation quality compared to state-of-the-art caching baselines. The authors demonstrate that MeanCache preserves both perceptual quality and content consistency under high acceleration. e.g. rare-word prompts remain consistent under 4× speedup for MeanCache, whereas prior methods exhibit severe drift. The key contributions are a new average-velocity perspective for caching and a principled stability-driven scheduling algorithm, together yielding a simple yet effective acceleration scheme.

**Strengths:**

* MeanCache presents a conceptually simple shift (using average vs instantaneous velocity) yet this new perspective is powerful. The paper explains this insight clearly.

* It achieves large acceleration on realistic large models while retaining high fidelity, outperforming prior caching schemes across image/video tasks.

* The use of the MeanFlow identity and JVP to bridge instantaneous and average velocity is well-founded.

* The trajectory-stability scheduling (peak-suppressed shortest path) is a novel, effective tool for deciding cache placement under budget.

* The method is training-free and was tested on diverse tasks (text-to-image, text-to-video), suggesting broad applicability.

**Weaknesses:**

* The method depends on approximating a future JVP from past states. As the authors note, the choice of interval K is “critical” and this is a trade-off between error and stability. If the model’s dynamics are highly non-linear, the approximation might degrade. The paper addresses this with scheduling, but it remains an approximation-dependent approach.

* There are several hyperparameters (cache span $K$, budget $\mathcal{B}$, peak-penalty $\gamma$). While the paper provides ablation studies, choosing these may be non-trivial in new settings.

* The scheduling step involves building a multigraph of possible cache edges and solving a shortest-path; the paper does not discuss this overhead.

**Questions:**

* How sensitive is the chosen cache schedule to the specific prompts or seeds used to build the stability map? In practice, do you compute a single schedule offline per model, or would it need adjustment at runtime for different inputs?

* If the diffusion model is updated (e.g. fine-tuned or improved), must the whole schedule and hyperparameters (like K spans) be recomputed? Is there a way to adapt the existing cache without re-running the full graph optimization?

* How much overhead is incurred when solving the shortest-path?

---

> ### Author Response · Authors · 2025-11-24
> **Response to Reviewer Egs2 (1/2)**
>
> > **W1: If the model's dynamics are highly non-linear, the approximation might degrade. The paper addresses this with scheduling, but it remains an approximation-dependent approach.**
>
> Thank you for the insightful comment. Cache-based acceleration methods inherently assume that the denoising trajectory does not exhibit extreme non-linearity over short intervals. In practice, however, instantaneous velocities can occasionally become less smooth or oscillatory, which amplifies accumulated approximation error. Our goal is not to eliminate this limitation completely, but to mitigate it as much as possible. The following points explain why MeanCache is able to effectively reduce accumulated errors:
>
> * **Smoother representation via average velocity.**
>   Average velocity captures interval-level behavior rather than instantaneous fluctuations, reducing local curvature and improving extrapolation stability.
>
> * **JVP encodes relationships rather than raw values.**
>     The cached JVP arises from the MeanFlow identity and characterizes how the average velocity relates to the instantaneous velocity. This relational quantity varies more smoothly and is more stable than the pointwise values of the velocity itself, making it a more reliable basis for extrapolation across multiple steps.
>
> * **Scheduling avoids unstable segments.**
>   The stability map identifies regions where the approximation becomes unreliable, and the peak-suppressed shortest-path selects spans only where the dynamics remain sufficiently smooth.
>
> Taken together, these factors allow MeanCache to remain robust despite relying on an approximation, even when parts of the trajectory exhibit non-linear or irregular behavior.
>
> ---
>
> > **W2: Hyperparameter selection may be difficult in new settings**
>
> Thank you for the comment. In practice, the hyperparameter tuning cost of MeanCache is relatively low.
>
> During preprocessing, only two lightweight parameters need to be provided: $\mathbf{K}$ and $\boldsymbol{\gamma}$. The cache span $\mathbf{K}$ simply requires a small candidate range (typically $\mathbf{K} \le 5$, as larger spans tend to introduce higher approximation error). In addition, our ablations show that the performance is insensitive to $\boldsymbol{\gamma}$, so no fine-grained tuning is necessary.
>
> During inference acceleration, the only parameter users need to choose is the budget $\mathbf{B}$, which provides a flexible control for balancing speed and quality without any task-specific tuning.
>
>
> ---
>
> > **W3 & Q4: How much overhead is incurred when solving the shortest-path?**
>
> Thank you for the question. The time cost of the Stability Map comes from two sources:\
> (1) running the original denoising trajectory, which depends on the model’s inherent inference time, and\
> (2) computing the error between the true and predicted average velocity (Eq. 13), which is consistently around 8.15 seconds per sample in our measurements.
>
> Belows are the time for a single sample on FLUX.1 with H100/80GB:
>
> | Stage                                        | Cost (seconds)                               | Impact on Inference Time |
> | -------------------------------------------- | -------------------------------------------- | ------------------------ |
> | **Stability Map**                            | 20.23 (includes 8.15s for error computation) | No                       |
> | **Peak-Suppressed Shortest-Path Scheduling** | < 0.02                                       | No                       |
> | **Model Inference**                          | 2.91x \~ 4.12x speedup                       | Yes                      |
>
> So for full overhead in our experiments(20-50 samples), it takes 6.7-16.8min to construct the stability map and less than 0.02s to solve the Peak-Suppressed shortest-path. Moreover, Stability Map construction is fully parallelizable across prompts; multi-GPU execution can greatly reduce the wall-clock time.

---

> ### Author Response · Authors · 2025-11-24
> **Response to Reviewer Egs2 (2/2)**
>
> > **Q1-1: How sensitive is the chosen cache schedule to the specific prompts or seeds used to build the stability map?**
>
> Thank you for the question. In practice, the stability map is not sensitive to the particular prompts or seeds used during its construction.
>
> Actually, our method is built using Out-Of-Domain(OOD) samples. Following TeaCache, T2V-CompBench (independent text-to-video dataset) is selected to construct the map, whereas the evaluation is conducted on text-to-image benchmarks. Despite this distribution gap, the schedule remains effective, indicating strong generalization.
>
> To further verify robustness, we used the FLUX.1 and evaluated MeanCache on three challenging sub-tasks while **reusing the same schedule** (i.e., without recomputing the stability map):
>
> *   **Rare Words:** prompts containing uncommon or extremely low-frequency words
> *   **Misspellings:** intentionally misspelled prompts
> *   **Text:** prompts requiring embedded textual content in the generated image
>
> The results below show that MeanCache consistently outperforms TaylorSeer, demonstrating that the schedule generalizes well even under difficult and out-of-distribution cases.
>
> | Method                | Sub-task     | LPIPS ↓  | SSIM ↑   | PSNR ↑    |
> | --------------------- | ------------ | -------- | -------- | --------- |
> | TaylorSeer (N=6, O=1) | Rare Words   | 0.47     | 0.61     | 16.70     |
> |                       | Misspellings | 0.47     | 0.63     | 17.43     |
> |                       | Text         | 0.46     | 0.57     | 15.01     |
> | **MeanCache (B=15)**  | Rare Words   | **0.27** | **0.78** | **21.47** |
> |                       | Misspellings | **0.19** | **0.84** | **24.75** |
> |                       | Text         | **0.16** | **0.82** | **21.99** |
>
> These results support that the cache schedule is highly stable and does not require per-input or per-task adjustment at runtime.
>
> ---
>
> > **Q1-2: In practice, do you compute a single schedule offline per model, or would it need adjustment at runtime for different inputs?**
>
> Yes, for a given $\mathbf{B}$ (determines how fast you expect the model run), we compute a static schedule offline and reuse it, keep it unchanged across all inputs, as our experiments show that this static prior is sufficient to provide stable performance across different inputs.
>
> ---
>
> > **Q2: If the diffusion model is updated (e.g., fine-tuned or improved), must the whole schedule and hyperparameters be recomputed? Is there a way to adapt the existing cache without re-running the full graph optimization?**
>
> Thank you for your insightful question. From our point of view, the current diffusion base models(Flux.1, QWen-Image, etc) are well-trained, their underlying foundation is stable and not easily affected by fine-tuning or similar modifications. So for typical fine-tuning (e.g., style or character adjustments), there is no need to recompute the cache scheduling rules. The denoising trajectory structure typically remains similar in these cases. However, if the fine-tuning is aimed at acceleration or involves full-scale training that leads to a significant shift in the underlying model, we recommend re-running. Currently, MeanCache does not support partial schedule updates, but incremental refinement of the stability map is an interesting direction we plan to explore in future work.

---

### Author Response · Authors · 2025-12-01

We thank the ACs, SACs, PCs, and all reviewers for their time and efforts in carefully reviewing our submission and for the many constructive comments and suggestions.

---

#### **1. Restating the Core Contribution**

The core contribution of this paper is a pivotal paradigm shift in caching methods: introducing an **average velocity** perspective to replace the traditional instantaneous velocity approach. By integrating this with our proposed trajectory-stability scheduling strategy (Stability Map), MeanCache effectively mitigates the cumulative error problem prevalent in existing caching methods. It achieves SOTA performance across multiple commercial-scale T2I and T2V models.

---

#### **2. Positive Feedback and Additional Work**

We are highly encouraged by the positive reception from the majority of reviewers. Reviewer Egs2 described our approach as "a conceptually simple shift... yet this new perspective is powerful," while Reviewers 4J2V and 5MTc deemed the motivation clear, the method practical, and the results strong.

Based on the insightful suggestions from the reviewers, we have completed extensive theoretical analysis and corresponding experiments during the rebuttal phase:

*   **Theoretical Analysis & Error Bounds:** We provided a complete theoretical analysis (highlighting the superiority of average velocity over instantaneous velocity in caching, and how the Stability Map avoids unstable segments) and supplemented it with a rigorous Error Bound analysis (Egs2-W1 and 5MTc-W1).
*   **Methodological Comparison:** We detailed the intrinsic differences between MeanCache and TaylorSeer (4J2V-W1), demonstrating the superiority of our approach.
*   **Cost, Overhead, and Efficiency:** We included a detailed analysis of offline costs (Egs2-W3 & Q4 and 4J2V-W2), provided an amortization analysis (5MTc-W3 & Q3), and reported the memory footprint (4J2V-Q3), demonstrating high efficiency in practical deployment.
*   **Generalization Experiments:** We verified the superior robustness and generalization capabilities of the Stability Map across cross-task (Egs2-Q1-1, 4J2V-Q2) and cross-resolution scenarios (5MTc-W2 & Q1) through comprehensive experiments.
*   **Implementation Details:** We addressed questions regarding hyperparameter selection (Egs2-W2), cache scheduling updates (Egs2-Q2), and refined mathematical notation and expressions (pV1g-W6).

---

#### **3. Clarifying a Key Misunderstanding**

Reviewer pV1g (the sole negative rating) raised a central concern:
> **“The main contributions of the paper come from implementing feature caching within the MeanFlow model ... Why caching in MeanFlow would be better than caching in flow models? This part is most confusing and concerning to me.”**

We wish to emphatically clarify a fundamental misunderstanding here:
> **MeanCache is NOT a “feature caching method” within the “MeanFlow” model.**

Specifically, while the MeanFlow theoretically demonstrates the superiority of average velocity over instantaneous velocity, current commercial-scale models are trained based on instantaneous velocity and lack corresponding MeanFlow versions. Therefore, effectively introducing the advantages of average velocity into the caching acceleration of existing Flow Matching (FM) models remains an unresolved core problem.

To address this challenge, we re-derived and constructed an average velocity perspective tailor-made for caching, introducing the novel "JVP Caching + Stability Map" acceleration framework for the first time. **This is a new method for the general "FM inference caching" problem, not "feature caching within a MeanFlow model."**

As noted above, the evaluations from Reviewers Egs2, 4J2V, and 5MTc indicate that they accurately understood our core contribution. Regrettably, Reviewer pV1g incorrectly assumed we were performing caching within a MeanFlow model. We elaborated on this in detail in our response (pV1g-W1 & Q1).

Although Reviewer pV1g has not responded further, this clear and direct clarification addresses the fundamental interpretation bias, and we believe it resolves the reviewer's core concern.

---

#### **4. Conclusion**

We believe that through the detailed responses and supplementary experiments in the rebuttal phase, we have fully addressed the queries of all reviewers. All revisions will be integrated into the final version of the paper. Once again, we thank everyone for their professionalism and constructive engagement throughout the review process.


Best regards,

Authors

---

### Meta-Review · Area_Chair_MgAx · 2025-12-24

**Summary:**

This paper proposes a training-free caching framework for efficient Flow Matching inference. In the first round, this paper received four reviews (6 6 6 2). One negative reviewer expressed concern about the values compared to MeanFlow and the lack of sufficient explanation and rigor in the mathematical formula. After rebuttal, the author provided additional intuitive explanations but still lacked some quantitative experiments (how vanilla MeanFlow (without caching) performs). Considering the advantages of performance and efficiency, the paper is recommended for acceptance.

**Reviewer Concerns:**

After rebuttal, the author provided additional intuitive explanations but still lacked some quantitative experiments (how vanilla MeanFlow (without caching) performs).

**Reviewer Scores:**

I believe the authors can partially address the issues raised, and the reviewer who with scores of 2 is inclined to raise his score.

---

### Decision · Program_Chairs · 2026-01-26

Accept (Poster)